# CRITERIA AND BIAS OF PARAMETERIZED LINEAR REGRESSION UNDER EDGE OF STABILITY REGIME

## ABSTRACT

Classical optimization theory requires a small step-size for gradient-based methods to converge. Nevertheless, recent findings Cohen et al. (2021) challenge the traditional idea by empirically demonstrating Gradient Descent (GD) converges even when the step-size $\eta$ exceeds the threshold of $2/L$, where $L$ is the global smooth constant. This is usually known as the *Edge of Stability* (EoS) phenomenon. A widely held belief suggests that an objective function with subquadratic growth plays an important role in incurring EoS. In this paper, we provide a more comprehensive answer by considering the task of finding linear interpolator $\boldsymbol{\beta} \in \mathbb{R}^d$ for regression with loss function $l(\cdot)$, where $\boldsymbol{\beta}$ admits parameterization as $\boldsymbol{\beta} = \boldsymbol{w}_+^2 - \boldsymbol{w}_-^2$. Contrary to the previous work that suggests a subquadratic $l$ is necessary for EoS, our novel finding reveals that EoS occurs even when $l$ is quadratic under proper conditions. This argument is made rigorous by both empirical and theoretical evidence, demonstrating the GD trajectory converges to a linear interpolator in a non-asymptotic way. Moreover, the model under quadratic $l$, also known as a depth-2 *diagonal linear network*, remains largely unexplored under the EoS regime. Our analysis then sheds some new light on the implicit bias of diagonal linear networks when a larger step-size is employed, enriching the understanding of EoS on more practical models.

## 1 INTRODUCTION

In the past decades, gradient-based optimization methods have become the main engine in the training of deep neural networks. These iterative methods provide efficient and scalable approaches for the minimization of large-scale loss functions. A key question that arises in the context is under *what* conditions, these algorithms are guaranteed to converge. Classical analysis of gradient descent (GD) answers the question by asserting that a small step-size should be employed to ensure convergence. To be precise, for the minimization of $L$-smooth objective functions, sufficient condition rules that step-size $\eta$ should never come across the critical threshold $2/L$ (or equivalently $L < 2/\eta$). This guarantees every GD iteration decreases the objective until it converges. As a result, the iterative algorithm can be viewed as a discretization of a continuous ODE called Gradient Flow (GF), and the corresponding convergence behavior is therefore referred to as the GF or stable regime.

Nevertheless, in the work of Cohen et al. (2021), it is observed, in the training process of certain learning models, GD and other gradient-based methods still converge even when the classical condition is violated, allowing for the use of much larger step-sizes. When this occurs, the objective does not exhibit the typical monotonic decrease, and the sharpness, defined as the largest eigenvalue of the objective function's Hessian matrix, frequently exceeds the threshold of $2/\eta$. Unlike the GF regime under small step-size, the GD trajectory often becomes violent, exhibiting oscillating and unstable behavior. Despite this, convergence is still achieved in the long run. This unconventional phenomenon is usually known as the Edge of Stability (EoS) regime. Similar results are also observed for algorithms including momentum methods or adaptive methods (Cohen et al., 2022).

In the recent several years, the new finding from Cohen et al. (2021) has pioneered many works to investigate the fascinating phenomena, from both empirical and theoretical aspects. Many theoretical works that attempt to explain the mechanism of EoS suggest that the subquadratic growth of the loss function plays a crucial role in causing EoS (Chen & Bruna, 2022; Ahn et al., 2022a). In particular,

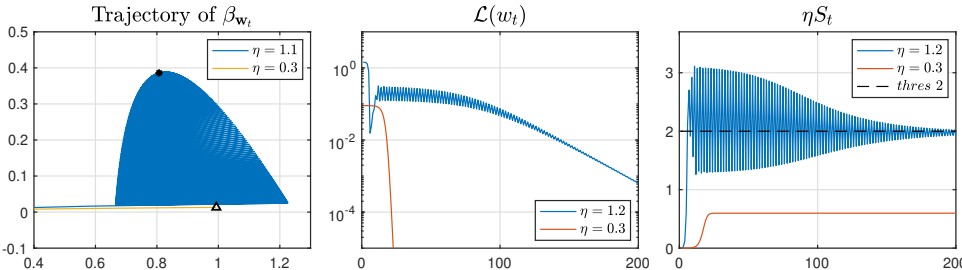

Figure 1: Comparison between EoS and GF regime, represented by blue and red lines, under parameterized linear regression in (1) with $l(a) = a^2/4$. The plots from left to right illustrate the trajectory of regression weight $\boldsymbol{\beta}_{\boldsymbol{w}_t}$ (star and triangle mark the stable points), the decrease of objective and $\eta S_t$, respectively, where $S_t$ is the sharpness at iteration $t$. EoS is featured by the $\eta S_t > 2$. Unlike previous assertions, we observe EoS *also* occurs with quadratic $l(a) = a^2/4$. Rest parameters: $\boldsymbol{x} = (1, 0.5)$, $y = 1$ and $\alpha = 0.01$.

Ahn et al. (2022a)[1] and its subsequent work Song & Yun (2023) considered the linear regression task of finding a vector $\boldsymbol{\beta}$ that interpolates single-point data $(\boldsymbol{x}, y)$ by minimizing the empirical risk $l(\langle \boldsymbol{x}, \boldsymbol{\beta} \rangle - y)$. The authors have rigorously shown that when $\boldsymbol{\beta}$ is allowed a parameterized form $\boldsymbol{\beta} = \boldsymbol{\beta}(\boldsymbol{u}, \boldsymbol{v})$, a loss $l(\cdot)$ with strictly subquadratic growth leads to the EoS phenomenon in running GD. When EoS occurs, the GD trajectory converges despite being highly unstable, oscillating almost symmetrically around zero along the primary axis.

In this work, we aim to challenge the notion that subquadratic loss is necessary for causing EoS by investigating the same regressional task. We specify that the weight vector $\boldsymbol{\beta}$ admits **quadratic parameterization** as $\boldsymbol{\beta} = \boldsymbol{w}_+^2 - \boldsymbol{w}_-^2$, which recovers and extends the setting of Ahn et al. (2022a). Our study presents both empirical and theoretical evidence to show that, when a **quadratic** loss $l(a) = a^2/4$ is employed, GD with large step-size can converge to a linear interpolator within the EoS regime, provided certain **conditions** are met. We believe this the *first* among existing works to suggest that quadratic loss function can trigger EoS and to characterize the convergence.

It is important to note that the model we consider is not merely an artificially designed landscape solely to induce EoS under strict conditions. This investigation is related not only to the community exploring the intriguing convergence under irregularly large step-size but also to the broader community tackling implicit bias of gradient methods: when $l(\cdot)$ is quadratic, our framework recovers the renowned model of depth-2 *diagonally linear networks*. A diagonal linear network captures the key features of deep networks while maintaining a simple structure, as each neuron connects to only one neuron in the next layer. Therefore, the study on the bias and generalization properties of this model has resulted in a rich line of important works in the past several years (Woodworth et al., 2020). While it has become well-studied when running GD with a classical step-size, it remains rather unexplored when it enters the EoS regime. We believe our work provides useful insights into the study of implicit bias of diagonal linear networks under large step-size by considering a simple but intuitive one-sample setting.

### 1.1 OUR CONTRIBUTION AND RELATION TO PREVIOUS WORKS

We discuss the scope and major contributions of our work below.

- We consider running GD with a large constant step-size to find linear interpolators that admit quadratic parameterization for the one-sample linear regression task in $\mathbb{R}^d$. This model is also called the depth-2 diagonal linear network. We show that empirically, convergence in the EoS regime is possible when $d > 1$ and the data does not constitute a degenerate case that can be reduced to a $d = 1$ setting.

- The above conditions are verified in a theoretical analysis, in which we show that the iteration of GD converges to a linear interpolator $\boldsymbol{\beta}_\infty$ under the EoS regime. We provide convergence

---

[1]To be exact, Ahn et al. (2022a) originally considered the model $(u, v) \mapsto l(uv)$ where $l$ is a loss function that grows subquadratically and did not reformulate it as a linear regression with parameterized weight vector. Instead, the formulation of regression setting was introduced in the subsequent Song & Yun (2023), and Ahn et al. (2022a)'s model is a special case of it. We combine the discussion here to give the audience a more complete picture.

analysis for two sub-regimes under EoS, i.e. $\mu\eta < 1$ and $\mu\eta > 1$ ($\mu$ is a scaling parameter related to the data $(\boldsymbol{x}, y)$), which exhibit different convergence behavior.

- In addition, we also characterize the generalization property for the implicit bias under the EoS regime by establishing upper bounds for $\|\boldsymbol{\beta}_\infty - \boldsymbol{\beta}^*\|$, where $\boldsymbol{\beta}^*$ is the given sparse prior.

- We also extend the one-sample results by empirically finding conditions in the more general $n$-sample case. This suggests that a non-degenerate overparameterized setting ($d > n$), is necessary for the EoS phenomenon.

It is important to emphasize that our findings should not be interpreted as overturning existing results. Rather, they improve and complement the existing understanding of what conditions lead to EoS for the parameterized linear regression with quadratic $l$ by identifying additional criteria. This does not contradict that non-quadratic property is necessary to incur EoS, because, despite $l$ being quadratic, parameterization $\boldsymbol{\beta} = \boldsymbol{w}_+^2 - \boldsymbol{w}_-^2$ ensures non-vanishing third-order derivative of the entire objective function, which is proven to be important in works like Damian et al. (2022).

We particularly highlight the significance of the proof under $\eta\mu > 1$. A brief explanation is provided here, with further details in Section 5. Our proof technique is highly related to the bifurcation analysis of discrete dynamical systems, which indicates a parameterized family of systems, as $w_{t+1} = f_a(w_t)$, display different asymptotic properties[2] if $a$ is fixed and takes different values. Some recent works including Chen et al. (2023) showed that such a system with fixed $a$ can describe GD trajectory for certain models, guaranteeing the convergence for these models, whereas running GD on our model corresponds to a system with *varying* $a$. This poses a more challenging task than ever, therefore our major innovation and difficulty is to show that under certain conditions, a phase transition will occur, such that the system travels on the phase diagram and finally becomes convergent.

## 1.2 RELATED WORKS

**Edge of Stability.** Although the phenomena of oscillation during the training process have been observed in several independent works (Xing et al., 2018; Lewkowycz et al., 2020; Jastrzebski et al., 2021), the name of Edge of Stability was first found in Cohen et al. (2021), which provided more formal definition and description. Among the theoretical exploration, some works attempt to find criteria that allow EoS to occur on general models (Ma et al., 2022; Damian et al., 2022; Ahn et al., 2022b; Arora et al., 2022). Nevertheless, these works do not provide very convincing arguments because they often incorporate demanding assumptions. An approach that is closer to this paper considers specific models and characterizes the convergence or implicit bias when EoS occurs, including (Chen & Bruna, 2022; Zhu et al., 2022; Ahn et al., 2022a). Recently, such analysis has been extended to more complicated and more practical models, for instance, logistic regression (Wu et al., 2024), parameterized linear regression (Song & Yun, 2023; Lu et al., 2023) and quadratic models (Chen et al., 2023). It is also worth mentioning that, instead of explaining the unstable convergence of EoS, some works including Li et al. (2022) studied how the sharpness grows during the early phases of the GD trajectory.

**Implicit bias of diagonal linear networks.** The diagonal linear network model, also called linear regression with quadratic parameterization, is one of the simplest deep network models that exhibit rich features and bias structure. Vaskevicius et al. (2019); Zhao et al. (2022); Gunasekar et al. (2017) were among the first works to explore the implicit bias when the weight admits a Hadamard parameterization $\boldsymbol{u} \odot \boldsymbol{v}$, which is provably equivalent to the quadratic parameterization $\boldsymbol{w}_+^2 - \boldsymbol{w}_-^2$. The seminal work of Woodworth et al. (2020) demonstrated that the scale of initialization decides the transition between rich and kernel regime, and also the recovery of sparse prior for diagonal linear networks. The subsequent works also considered topics such as the connection to Mirror Descent (Gunasekar et al., 2021; Azulay et al., 2021), stochastic GD (Pesme et al., 2021), and limiting initialization (Pesme & Flammarion, 2023). Recently, several papers Nacson et al. (2022); Even et al. (2023); Andriushchenko et al. (2023) attempted to address the bias of (S)GD under the large step-size regime, nevertheless, they failed to establish the convergence when EoS occurs.

---

[2]By this we mean the system being convergent to stable point or stable periodic orbits or becoming chaotic or divergent. Only convergence to the stable point of 0 is the case we want to show.

## 2 PRELIMINARY AND SETUP

**Notations.**  We introduce the notations and conventions used throughout the whole paper. Scalars are represented by the simple lowercase letters. We use bold capital like $A$ and lowercase letters like $v$ to denote matrix and vector variables. For vector $v$, let $\|v\|_2$ denote its $l_2$-norm and $v^p$ denote the coordinate-wise $p$-power. Also, for two vectors $u, v$ of the same dimension, let $u \odot v$ denote their coordinate-wise product. Let $I$ be the identity matrix. For any integer $n > 0$, let $[n] := \{1, \dots, n\}$. We use asymptotic notations $O(\cdot)$, $\Omega(\cdot)$ and $\Theta(\cdot)$ in their standard meanings.

### 2.1 MODEL AND ALGORITHM

**Regression with quadratic parameterization.**  We consider the linear regression task on *single* data point $(x, y)$, where $x \in \mathbb{R}^d$ and $y \in \mathbb{R}$. Following Ahn et al. (2022a); Song & Yun (2023); Lu et al. (2023), it is instructive to study the one-sample setting because it (1) is sufficiently simple to analyze and (2) demonstrates the key feature under the Edge of Stability regime. This setup is overparameterized when $d \geq 2$ and therefore admits infinitely many linear interpolators $\beta$ satisfying $\langle x, \beta \rangle = y$. We aim to find one of these linear interpolators by minimizing the empirical risk:

$$\mathcal{L}(\beta) = l(\langle x, \beta \rangle - y), \tag{1}$$

where $l(\cdot)$ is convex, even, and at least twice-differentiable, with the minimum at $l(0) = 0$.

We consider the model where the regression vector $\beta$ admits a *quadratic* parameterization

$$\beta := \beta_w = w_+^2 - w_-^2, \quad w_\pm \in \mathbb{R}^d, \quad w = \begin{bmatrix} w_+ \\ w_- \end{bmatrix} \qquad \text{(Quadratic Parameterization)}$$

where $w$ is the trainable variable. With an abuse of notation we write $\mathcal{L}(w) := \mathcal{L}(\beta_w)$. In particular, when $l(\cdot)$ is quadratic, the model is also called the diagonal linear network, well-studied under a small step-size regime in the past several years (Woodworth et al., 2020; Gunasekar et al., 2021).

We minimize the loss function $\mathcal{L}(w)$ by running GD with constant step-size $\eta$: for any $t \in \mathbb{N}$, it formalizes the following iteration

$$w_{t+1} = w_t - \eta \nabla_w \mathcal{L}(w_t). \tag{2}$$

Unpacking the definition in (1), the gradient of the loss function can be written as

$$\nabla_w \mathcal{L}(w) = 2l'(r(w)) \cdot \begin{bmatrix} +x \odot w_+ \\ -x \odot w_- \end{bmatrix}$$

where $r(w) = \langle \beta_w, x \rangle - y$ is referred as the *residual* at $w$ on sample $(x, y)$. In particular, if $l(a) = \frac{a^2}{4}$, its derivatives are simply as $2l'(r(w)) = r(w)$ and $2l''(r(w)) = 1$.

## 3 EoS UNDER QUADRATIC LOSS: AN EMPIRICAL STUDY

In this section, we empirically investigate the EoS convergence of GD on the model in (1) when the $\beta$ admits Quadratic Parameterization. Rigorously, we define EoS as the phenomena that the sharpness $S_t := \lambda_{\max}(\nabla^2 \mathcal{L}(w_t))$ crosses the threshold of $2/\eta$ for some $t$. In particular, we are interested in finding conditions for it to admit EoS when $l$ is a quadratic function, which has been less explored in the existing literature.

We briefly discuss the result from Ahn et al. (2022a) and its relation to our model. The authors considered $(u, v) \mapsto l(uv)$, which can be regarded as a special case of our model when $d = 1$, $x = 1$ and $y = 0$ due to linear transformation $(u, v) = (w_+ + w_-, w_+ - w_-)$ that remains invariant under GD. The authors proved the *necessary* condition for the model to admit EoS is that the loss function is subquadratic, i.e. there exists $\beta > 0$ such that $\frac{l'(a)}{a} \leq 1 - \Theta(|a|^\beta)$ when $a$ is small [3].

However, we deepen the understanding and provide a *sufficient* condition by empirically testing under which conditions EoS occurs even with $l(a) = a^2/4$ when we focus on the one-sample model in (1). The message is stated in the following claim.

---

[3]The subsequent work Song & Yun (2023) extended the result to general $d$, still requiring $l$ to be sub-quadratic unless non-linear activation is employed. Therefore it is not comparable to this result.

**Claim 1.** *Consider the one-sample risk minimization task in (1) with Quadratic Parameterization. For GD, it is sufficient for EoS to occur under properly chosen constant step-size with a quadratic loss $l(s) = s^2/4$ when the following conditions are satisfied: (1) $d \geq 2$, (2) $y \neq 0$ and (3) $\boldsymbol{x} = (x_1, \ldots, x_d)$ is not degenerated, i.e. $x_i \neq 0$ for any $i$ and there exists at least a pair $x_i \neq x_j$.*

We introduce the experimental configurations. For data generation, we sample $\boldsymbol{x} \sim \mathcal{N}(0, \boldsymbol{I}_d)$ and $y = \langle \boldsymbol{x}, \boldsymbol{\beta}^* \rangle$, where $\boldsymbol{\beta}^* \sim \text{Unif}(\{\beta \in \frac{1}{\sqrt{k}}\{0, \pm 1\}^d : \|\boldsymbol{\beta}^*\|_0 = k\})$. We use standard *scaling* initialization $\boldsymbol{w}_{\pm,0} = \alpha \boldsymbol{1}_d$ where $\alpha > 0$ is a factor. We choose the sparse prior and the scaling initialization because they are important in characterizing the implicit bias of GD under the GF regime, and hence make our results comparable to the results on diagonal linear networks like Woodworth et al. (2020).

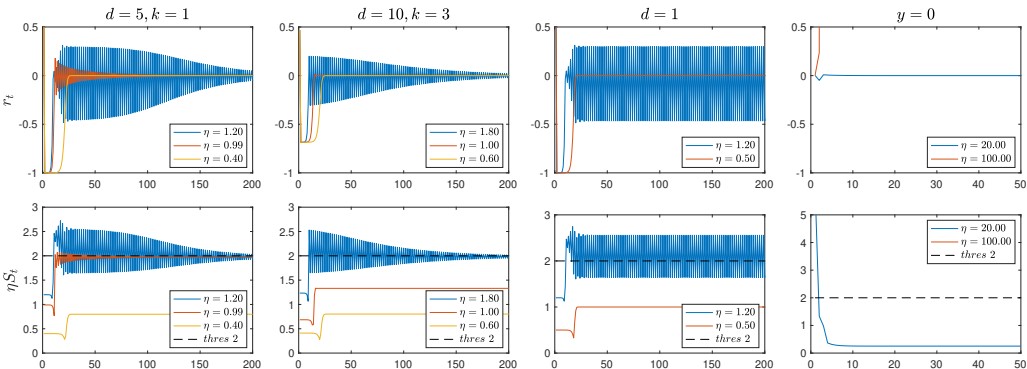

Figure 2: Empirical verification for the Claim 1. In the left two columns of plots, we run with configurations that obey Claim 1 and EoS occurs if we increase step-size. In contrast, we set $d = 1$ in the third column and $y = 0$ in the fourth column, under these settings GD becomes divergent without triggering EoS when we increase the step-size. Note that we use a modified initialization $\boldsymbol{w}_{0,+} = 2\alpha\boldsymbol{1}$, $\boldsymbol{w}_{0,-} = \alpha\boldsymbol{1}$ in the last column ($y = 0$), otherwise the $r_t = 0$ for any $t$ under the original initialization.

Before presenting our empirical evidence, we first explain the major empirical difference between EoS and GF regime via Figure 1. We characterize the convergence of GD via the residual function $r_t = r(\boldsymbol{w}_t)$ instead of loss $\mathcal{L}(\boldsymbol{w}_t)$ because the latter does not reflect the sign change. For our setting, the GF regime is featured by the monotonically decreasing of the $|r_t|$ until reaching 0. Besides, $r_t$ remains negative and the sharpness is under the threshold $2/\eta$ for any $t$. On the contrary, in the EoS regime, $r_t$ oscillates and changes its sign as $r_t r_{t+1} < 0$ holds for any $t$ large enough. Also, the sharpness first exceeds $2/\eta$ and decreases until it comes below the threshold. This is similar to the EoS behavior described in Ahn et al. (2022a). Nevertheless, the major difference between the GF regime and Ahn et al. (2022a) is, that the envelope of $r_t$ does not necessarily shrink under the EoS regime (which we will discuss later). Another distinction from Ahn et al. (2022a) is that the sharpness also oscillates.

We proceed by empirically justifying Claim 1. In Figure 2 we test if each one of the three conditions in Claim 1 is relaxed, EoS does not occur and GD diverges when we increase the step-size away from the GF regime. We intuitively explain why these conditions are necessary: the importance of $d > 1$ is predicted by the result of Ahn et al. (2022a). If (3) does not hold, e.g. $\boldsymbol{x} = x\boldsymbol{1}_d$ for some $x \neq 0$. The degenerate case reduces to the $d = 1$ setting and fails as violating (1).

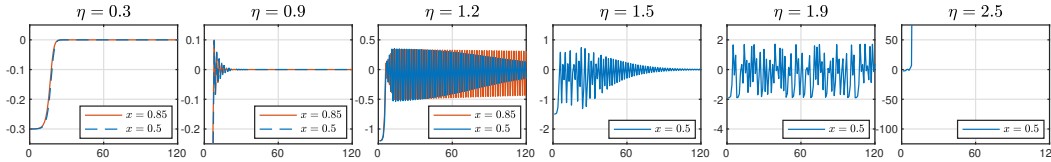

Figure 3: Influence of $\eta$ and different asymptotic properties of $r_t$ along GD trajectory. When we increase the step-size, it displays, from left to right, GF regime, different subregimes of EoS, chaos, and divergence. In particular, when $x$ is larger than some threshold (see Theorem 2 for details), GD does not converge when $\mu\eta > 1$. Parameter configuration: $\mu = 1$, $\alpha = 0.01$.

Now suppose all the conditions hold, we further investigate how the choice of step-size, scaling initialization, and other parameters affect the GD trajectory in the EoS regime. We examine how the scale of initialization and step-size might affect the oscillation, sparsity of solution, and the generalization property by testing on the case of $d = 2$ and 1-sparse prior $\boldsymbol{\beta}^* = (\mu, 0)$. The specific setting allows us to characterize it in a more qualitative manner and can be directly compared with theoretical analysis in the next section.

**Effect of step-size and types of oscillation.** The importance of step-size is not restricted to deciding EoS or GF regime, as illustrated Figure 3. We already describe the transition from GF to EoS and will focus on different *subregimes* of EoS. Another phase transition takes place at $\mu\eta$: if $\mu\eta < 1$, despite oscillating, the envelope of $r_t$ monotonically shrinks as $|r_{t+2}| < |r_t|$ after the initial phase; on the contrary, if $\mu\eta \in (1, \theta)$ ($\theta$ is a constant between (1,2)), the end of the initial phase does not mark the beginning of contraction. Instead, the envelope will expand un-

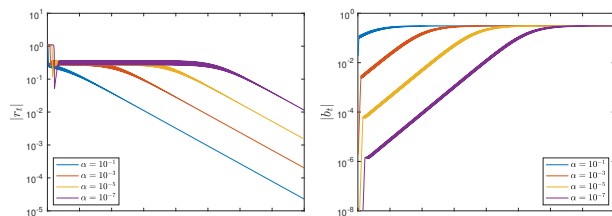

Figure 4: $\alpha$ decides the length of the intermediate phase in $\eta\mu > 1$: the gap between the start of oscillation $t_0$ and the start of convergence $t$ is proportional to $\log(1/\alpha)$. This is because in the intermediate phase, $r_t$ remains roughly as a constant and causes $b_t$ to increase almost linearly from the scale of $\alpha^{\Theta(1)}$ to $O(1)$. We use $x = 0.5$, $\eta = 1.1$ and $\mu = 1$.

til it saturates and reaches a 2-periodic orbit. This intermediate phase will maintain until another phase transition to the convergence phase occurs. Besides, $\alpha$ decides the length of the intermediate phase under the $\eta\mu > 1$ regime: the gap between the start of oscillation $t_0$ and the start of convergence $t$[4] empirically obeys $t - t_0 \propto \log(1/\alpha)$, as in Figure 4. Moreover, when $\eta$ is increased above $\theta$, $r_t$ will reach the orbit of a higher period during the intermediate phase. If we further increase $\eta$, the trajectory will finally become chaotic or divergent.

**Effect of $\alpha$, sparsity and generalization error.** We also care about how $\alpha$ decides the error $\|\boldsymbol{\beta}_\infty - \boldsymbol{\beta}^*\|^2$ where $\boldsymbol{\beta}_\infty$ is the limit of $\boldsymbol{\beta}_{\boldsymbol{w}_t}$, as in Figure 5. We focus on the generalization error under the EoS regime. Similar to the above discussion, it displays different behavior depending on $\eta\mu < 1$ or not. If $\eta\mu < 1$, $\|\boldsymbol{\beta}_\infty - \boldsymbol{\beta}^*\|$ will decrease almost linearly in $\alpha$ and recovers the sparse solution if $\alpha$ takes the limit of 0. On the contrary, when $\eta\mu > 1$, the error will be decided by two quantities:

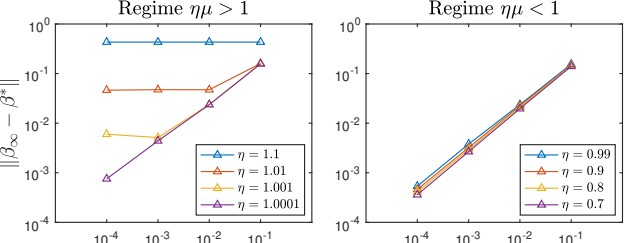

Figure 5: Relationship between error $\|\boldsymbol{\beta}_\infty - \boldsymbol{\beta}^*\|$, $\alpha$ and $\eta$: the $x$-axis is $\alpha$ and $y$-axis is the error. The left plot characterizes the error under $\mu\eta > 1$ and the right plot is for regime $\mu\eta < 1$. Rest parameters: $x = 0.5$, $\mu = 1$. The $x$-axis of both plots are in $\alpha$.

with $\alpha^2 \ll \mu\eta - 1$, the error $\|\boldsymbol{\beta}_\infty - \boldsymbol{\beta}^*\|$ is solely decided by $\eta$ regardless of the choice $\alpha$ and does not recover the sparse solution even when $\alpha \to 0$.

**Extension to multi-sample setting.** Our previous investigation focuses on the single-sample setting. Nevertheless, we believe that it is also very important to conduct an empirical study of the more general multiple data points case. Under this setting, the dataset $\{(\boldsymbol{x}_i, y_i)\}_{i=1}^n$ has $n$ data point, where $\boldsymbol{x}_i \in \mathbb{R}^d$ and $y_i = \langle \boldsymbol{x}_i, \boldsymbol{\beta}^* \rangle$. We aim at finding one linear interpolator by running GD over the following empirical risk $\mathcal{L}(\boldsymbol{w}) = \frac{1}{n} \sum_{i=1}^n l(\langle \boldsymbol{x}_i, \boldsymbol{\beta}_{\boldsymbol{w}} \rangle - y_i)$. Also, we use a similar experimental configuration for sparse prior $\boldsymbol{\beta}^*$ and initialization $\boldsymbol{w}_{0,\pm}$ as in the one-sample case, and the data is generated as $\boldsymbol{x}_i \sim \mathcal{N}(0, \mathbf{1}_d)$ and $y_i = \langle \boldsymbol{x}_i, \boldsymbol{\beta}^* \rangle$.

When a quadratic loss $l(\cdot)$ is employed, our empirical results suggest that the following two assumptions are important: (1) the setting is overparameterized, i.e. $d \geq n$ and (2). the setting is not degenerate. This is illustrated in Figure 6. In particular, it should be noticed that the overparame-

---

[4]A more detailed discussion of these quantities are reflected in Theorem 2 in the next section.

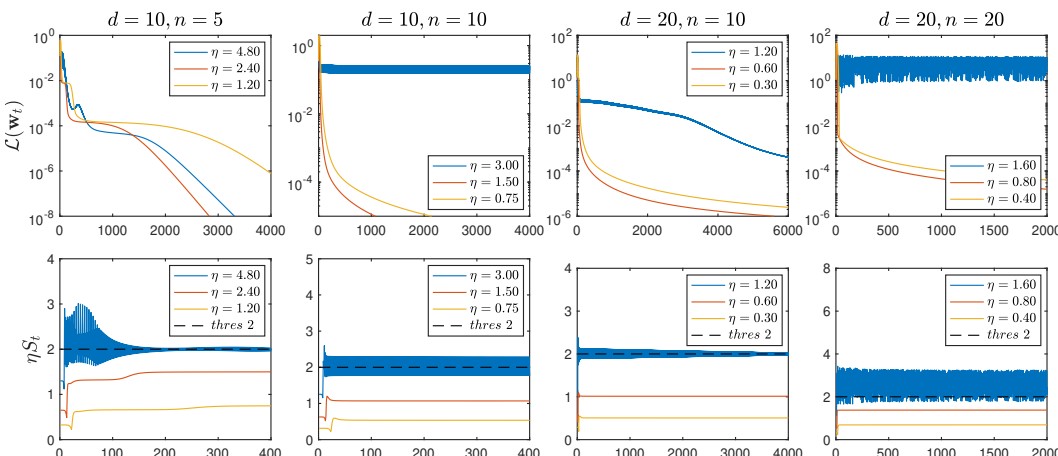

Figure 6: Empirical verification for the necessity of overparameterization under the multi-sample case. We plot the loss function $\mathcal{L}(\boldsymbol{w}_t)$ and the sharpness of GD when it admits more than one sample. When the model is overparameterized ($d > n$), EoS occurs when we increase the step-size. Otherwise, even with $d = n$, EoS does not occur and GD becomes unconvergent. Rest parameters: $k = 3$ and $\alpha = 0.01$.

terized condition $d \geq n$ reduces to $d \geq 2$ in the one-sample case, which is exactly condition (1) in Claim 1.

## 4 THEORETICAL ANALYSIS: BIAS UNDER ONE-SAMPLE CASE

Motivated by the empirical observations in Section 3, in this section, we aim to provide a theoretical explanation for the convergence of GD under the EoS regime when the loss function is quadratic. Moreover, we characterize its generalization error and compare it with existing results on diagonal linear networks under the GF regime. We begin by presenting the assumptions.

**Assumption 1.** *We make the following assumptions on (1) with Quadratic Parameterization:*

*(1). Suppose $d = 2$. Let $\boldsymbol{\beta}^*$ be 1-sparse as $\boldsymbol{\beta}^* = (\mu, 0) \in \mathbb{R}^d$ with $\mu \neq 0$. The data point $(\boldsymbol{x}, y)$ satisfies $\boldsymbol{x} = (1, x)$ and $y = \langle \boldsymbol{\beta}^*, \boldsymbol{x} \rangle = \mu$.*

*(2). The loss function $l(\cdot)$ is quadratic, i.e. $l(s) = s^2/4$;*

*(3). The initialization is set to be $\boldsymbol{w}_{0,\pm} = \alpha \mathbf{1}$ with $\alpha > 0$.*

We briefly discuss the motivation behind these assumptions. We choose the 1-sparse prior and the scaling initialization because they are important in characterizing the implicit bias for GD for the diagonal linear model, as mentioned in Section 3. For the choice of $\boldsymbol{x}$, we remark that the form $\boldsymbol{x} = (1, x)$ does not compromise the generality and recovers any input vector in $\mathbb{R}^2$ through rescaling. The rest conditions are required by Claim 1 and are therefore necessary for ensuring EoS under quadratic loss.

We present theorems to characterize the convergence of GD when the model we consider has dimension $d = 2$. It should be remarked that this is the *simplest* setting in which EoS occurs with a quadratic $l(\cdot)$. We provide a theoretical analysis to show that GD will converge to a linear interpolator under the EoS regime by discussing two cases depending on the choice of step-size $\eta$.

**Theorem 1.** *Suppose Assumption 1 and change of sign $r_t r_{t+1}$ occurs for any $t$ larger than some integer $t_0$. Let $\eta\mu \in (0, 1)$ and $\alpha^2 \leq O(1)$, then the GD iteration in (2) converges with a linear rate to the limit $\boldsymbol{\beta}_\infty$ as*

$$|\langle \boldsymbol{\beta}_{\boldsymbol{w}_t} - \boldsymbol{\beta}_\infty, \boldsymbol{x} \rangle| \leq C_1 \cdot e^{-\Theta(\mu\eta) \cdot (t - t_0)} \cdot |\langle \boldsymbol{\beta}_{\boldsymbol{w}_{t_0}} - \boldsymbol{\beta}_\infty, \boldsymbol{x} \rangle|.$$

*Moreover, $\|\boldsymbol{\beta}_\infty - \boldsymbol{\beta}^*\| \leq O(\alpha^{C_2})$. $C_1, C_2 > 0$ are some constants.*

**Theorem 2.** *Suppose Assumption 1. Let $\eta\mu \in (1, \min\{\frac{3\sqrt{2}-2}{2}, 1 + 1/(4\mathcal{C})\})$ and $\alpha^2 \ll \mu\eta - 1$, where $\mathcal{C}$ is a certain universal constant. If $x \in (-\frac{1}{\mu\eta}, \frac{1}{\mu\eta}) \setminus \{0\}$ holds, then there exists some $t$ such*

*that, the GD iteration in (2) converges with a linear rate to the limit $\boldsymbol{\beta}_\infty$ as*

$$|\langle \boldsymbol{\beta}_{\boldsymbol{w}_t} - \boldsymbol{\beta}_\infty, \boldsymbol{x}\rangle| \leq C_3 \cdot e^{-\Theta(\mu\eta-1)\cdot(t-\mathfrak{t})} \cdot |\langle \boldsymbol{\beta}_{\boldsymbol{w}_\mathfrak{t}} - \boldsymbol{\beta}_\infty, \boldsymbol{x}\rangle|.$$

*Moreover, $\|\boldsymbol{\beta}_\infty - \boldsymbol{\beta}^*\| \leq \mathcal{C} \cdot (\mu\eta - 1)$. $C_3 > 0$ are some constants.*

The above theorems indicate that under both $\mu\eta \in (0,1)$ and $\mu\eta \in (1, \frac{3\sqrt{2}-2}{2})$, if EoS occurs, we can establish linear convergence of $\boldsymbol{\beta}_{\boldsymbol{w}_t}$ to their respective limits $\boldsymbol{\beta}_\infty$, which are linear interpolators for the one-sample $(\boldsymbol{x}, y)$. Nevertheless, they are different in many perspectives, as whether $\eta\mu$ exceeds 1 decides some key distinct features of the oscillation trajectory. We will explain in the following remarks.

First, in the first theorem we require that a change of sign occurs. This is because, when $\eta\mu < 1$, GD could enter either EoS or GF regime depending on the exact value of $\eta$. Instead of identifying an exact threshold between GF and EoS (which can be very hard and purely technical), we simply employ this condition in Theorem 1 to rule out the possible choices of $\eta$ that lead to the GF regime and focus on the ones that lead to oscillating EoS regime.

Second, despite both $t_0$ and $\mathfrak{t}$ being markers for the beginning of linear convergence, they differ significantly in nature. As explained in Section 3, under both conditions, GD experiences a short initial phase, and its end is marked by $t_0$, which consequently also marks the beginning of convergence in $\mu\eta < 1$. On the contrary, in $\eta\mu > 1$, the envelope is not guaranteed to shrink after $t_0$. And the phase transition to the third convergence phase is marked by $\mathfrak{t}$ if the assumption $|x| \in (0, \frac{1}{\mu\eta})$ is met.

Moreover, we discuss generalization error by bounding $\|\boldsymbol{\beta}_\infty - \boldsymbol{\beta}^*\|$. When $\eta\mu \leq 1$, the generalization bound at infinity $\|\boldsymbol{\beta}_\infty - \boldsymbol{\beta}^*\|$ is dominated by $\alpha^{\Theta(1)}$, which vanishes and recovers the sparse prior as $\alpha$ approches 0, similar to the implicit-bias analysis under GF regime in Woodworth et al. (2020). Nevertheless, when $\eta\mu$ exceeds 1, Theorem 2 suggests that the error will depend on gap $\mu\eta - 1$ when $\alpha$ is small, and therefore does not recover the sparse solution when $\alpha \to 0$. This theoretical analysis thus matches the experimental observations in Section 3.

Lastly, we remark that in Theorem 2, the choice of upper bound $\frac{3\sqrt{2}-2}{2} \approx 1.12$ for $\eta\mu$ is purely artificial and comes primarily from the technical reasons. As discussed in Section 3, if $\eta$ is further increased from it, the envelope of $r_t$ will reach an orbit with periodicity longer than 2. The transition from 2-cycle to longer cycle does not occur at $\eta\mu = \frac{3\sqrt{2}-2}{2}$. We choose this value because it can be proved that it guarantees a 2-orbit and hence simplifies the proof. We next present a result showing the convergence when $\eta\mu \in (1, \frac{3\sqrt{2}-2}{2})$ is not necessary, though it comes at the cost of very restrictive assumptions.

**Proposition 1.** *Suppose Assumption 1. Let $\eta\mu > 1$ and $x \in (-\frac{1}{\mu\eta}, \frac{1}{\mu\eta}) \setminus \{0\}$. If the GD iteration is not diverging or becoming chaotic, then it converges to a linear interpolator $\boldsymbol{\beta}_\infty$ when $t$ goes to infinity.*

We provide an overview of our proof technique in the next section, with a toy example to better explain our method. For the complete proof, please refer to the Appendix. We hope our analysis can be extended to any $d \geq 2$ in future works. However, we emphasize this does not diminish the significance of our analysis, since the settings under different $d$ share EoS patterns and asymptotic behavior, and the technical barrier comes from linear algebra limitations.

## 5 PROOF OVERVIEW

In this section, we discuss the idea and technique used in our proof. In specific, we introduce a toy model–a nonlinear system with unstable convergent dynamics, similar to the behavior of EoS.

### 5.1 A TOY MODEL

We consider variable $r_t \in \mathbb{R}$ and the following iteration of a nonlinear system with initialization $r_0$:

$$r_{t+1} = -(1 - \alpha_t) \cdot r_t - \beta_t \cdot r_t^2, \tag{3}$$

where $\alpha_t, \beta_t \in \mathbb{R}$ are time-dependent inputs and we will assume $|\alpha_t| \leq 1$ for any $t \in \mathbb{N}$. We are concerned about the convergence of $|r_t|$ to zero when $t$ goes to infinity. Empirical results in Figure 7 indicate that (3) displays different regimes depending whether $\alpha_t < 0$ or $\alpha_t > 0$.

**Constant $\alpha$.** We begin with a simple setting where $\alpha_t, \beta_t$ are both constants, as $\alpha$ and $\beta > 0$. When $\alpha > 0$, $r_t$ alternates signs in each iteration and its envelope decreases, displaying damped oscillation. The convergence is formally established in the subsequent lemma. We remark that these lemmas are employed to explain our proof ideas, so we use proper assumptions to simplify their proof.

**Lemma 1.** *Let $\beta = 1$ and $\alpha \in (0, 1)$. Suppose with some proper initialization $r_0$, $r_t r_{t+1} < 0$ holds for any $t \in \mathbb{N}$. Then for any $r_t$ with $r_t < 0$, it satisfies $|r_{t+2}| < (1-\alpha)^2 \cdot |r_t|$ and hence the iteration admits limit $\lim_{t\to\infty} r_t = 0$.*

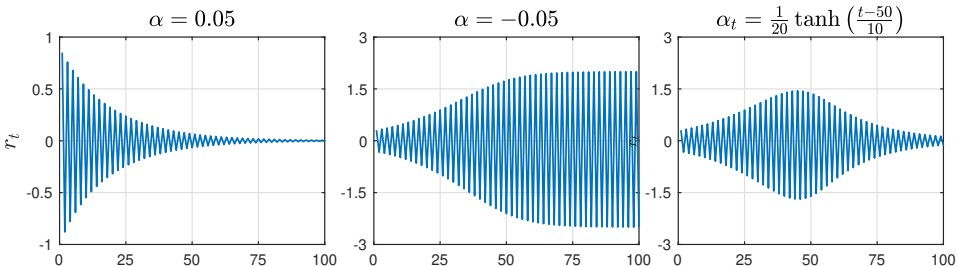

Figure 7: Toy model dynamics of $r_t$ in (3) under different regimes. The left and the middle plots utilize constant $\alpha$ with different signs and correspond to the *oscillating contracting* and the *expanding* regimes. The right plot uses varying $\alpha_t = \frac{1}{20} \cdot \tanh(\frac{t-50}{10})$ , which exhibits a phase transition when $\alpha_t$ crosses 0. In all the plots we use $\beta = 0.1$ and $r_0 = -0.3$.

Instead, when $\alpha < 0$, the iteration does not necessarily converge to zero. Nevertheless, with a proper initialization $r_0$, the sequence of $r_t$ oscillates and its envelope expands until reaching a certain value, as shown in Figure 7. The envelope-expanding behavior is very similar to the intermediate phase of GD under step-size $\eta\mu > 1$ mentioned in Section 3. This and the limit points are characterized in the following lemma.

**Lemma 2.** *Let $\beta = 1$ and $\alpha = -a$, with $a \in (0, 1)$ and define*

$$r_+ = \tfrac{1}{2}(\sqrt{a^2 + 4a} - a) > 0, \qquad r_- = \tfrac{1}{2}(-\sqrt{a^2 + 4a} - a) < 0.$$

*Then for any $r_t$ in (3) with $r_t \in [r_-, r_+]$, it holds that $|r_{t+2}| > |r_t|$ and $r_t r_{t+1} < 0$. Also, consider the subsequence of $r_t$'s being positive and negative. Then the two subsequences admit limit as $r_+$ and $r_-$.*

**Phase transition with varying input.** We have shown when $\alpha_t$ is a constant, it demonstrates different behavior when $\alpha$ admits different signs. Nevertheless, we now consider the case where a varying $\alpha_t$ is allowed. Especially, we initialize $\alpha_t$ as negative in the beginning phases and gradually increase $\alpha_t$ to become positive. Intuitively, the iteration will display a *phase transition* from the self-limiting regime to the *damping oscillation* regime when $\alpha_t$ changes signs. This is confirmed by running examples under such inputs, as illustrated in Figure 7.

### 5.2 PROOF OVERVIEW OF MAIN THEOREMS

We now use the idea from toy model analysis to illustrate our proving strategy for theorems in Section 4. We first prove that GD iteration on our model is equivalent to an iteration of a quadruplet as in the next lemma. This allows us to tackle an equivalent and much simpler iteration by avoiding matrix-vector products.

**Lemma 3** (Informal). *Running GD on the model in (1) with Quadratic Parameterization corresponds the following iteration of quadruplet $(a_t, a'_t, b_t, b'_t) \in \mathbb{R}^4$:*

$$a_{t+1} = (1 - \eta r_t)^2 \cdot a_t, \qquad b_{t+1} = (1 - x \cdot \eta r_t)^2 \cdot b_t,$$
$$a'_{t+1} = (1 + \eta r_t)^2 \cdot a'_t, \qquad b'_{t+1} = (1 + x \cdot \eta r_t)^2 \cdot b'_t,$$

*where we have $r_t = (1 + w^2)(a_t - a'_t + x \cdot (b_t - b'_t)) - \mu$.*

Still, dealing with four variables remains a challenging task. Aided by the empirical observation (see Figure 8) that $a'_t$ and $b'_t$ are almost unchanged from the initial position throughout the whole

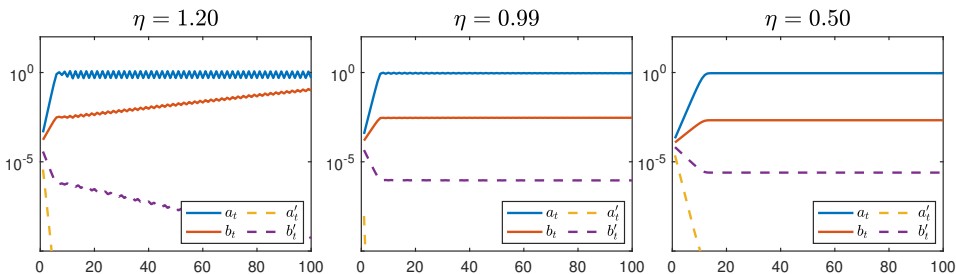

Figure 8: Equivalent dynamics of quadruplet $(a_t, a_t', b_t, b_t')$ in Lemma 3. We compare EoS regime ($\eta = 1.2$) and GF regime ($\eta = 0.5$) in terms of sharpness, trajectory and convergence of $|r_t|$, where $\Delta a_t = a_t - a_t'$ and $\Delta b_t = b_t - b_t'$. In the first plot, we only picture the trajectory under $\eta = 1.2$ to show that $a_t', b_t'$ remain almost fixed throughout the whole period.

time range, we can further simply the dynamics by fixing the two variables. This gives an iteration with only $(a_t, b_t)$.

It is not hard to observe that to establish convergence, it suffices to show that the residual term $r_t$ converges to zero. We observe that $r_t$ admits an update similar to the toy model iteration in (3):

$$r_{t+1} = -(1 - \alpha_t) \cdot r_t - \beta_t \cdot r_t^2,$$

where $\alpha_t$ and $\beta_t$'s are time-dependent variables (definition see Appendix A.1). The case $\mu\eta < 1$ in Theorem 1 corresponds to $\alpha_t > 0$ throughout the time and the convergence can be proven using an argument similar to Lemma 1. The more difficult one is the setting $\eta\mu > 1$ considered in Theorem 2, which implies $\alpha_t < 0$ and does not lead to convergence. Nevertheless, by a contradiction argument we show that if $|x| < \frac{1}{\mu\eta}$ is true, $\alpha_t$ will decrease in an oscillatory style until it becomes negative. The change of signs leads to a phase transition, allowing us to establish convergence. To the best of our knowledge, previous works including Chen et al. (2023) majorly tackled the settings similar to $\mu\eta < 1$, and hence $\alpha_t > 0$ holds. We believe our result is innovative because we are the *first* to show convergence for the case similar to $\mu\eta > 1$ that requires to show a phase transition does occur.

## 6 CONCLUSION

In this paper, we consider the task of finding interpolators for the linear regression with quadratic parameterization and study the convergence of constant step-size GD under the large step-size regime. In particular, we focus on the non-trivial question of whether a quadratic loss can trigger the Edge of Stability (EoS) phenomena or not, which seems unlikely from previous literature. Nevertheless, we show through both empirical and theoretical aspects that, when some certain condition is satisfied, EoS indeed occurs given quadratic loss. We hope this novel result takes a step further toward understanding the intriguing phenomena of unstable convergence.

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

# Appendix: Proofs and Supplementary Materials

## A  PROOF OF THE MAIN THEOREMS

In this section, we present the proof of our major result, i.e. Theorem 1, Theorem 2, and Proposition 1. The proof is very complicated and consists of several different parts. To improve the readability, we provide a sketch and an outline before proceeding to it.

**The sketch and outline of the proof.**  The major point of our theorems is to prove that $\boldsymbol{\beta}_{\boldsymbol{w}_t} = \boldsymbol{w}_{t,+}^2 - \boldsymbol{w}_{t,-}^2$ converges in the EoS regime to a linear interpolator for the sample $(\boldsymbol{x}, y)$, where $\boldsymbol{x} = (1, x)$ and $y = \mu$. This is formally stated as

$$\lim_{t\to\infty} \langle \boldsymbol{\beta}_{\boldsymbol{w}_t}, \boldsymbol{x} \rangle = \mu.$$

This is equivalent to saying that the residual function $r_t = \langle \boldsymbol{\beta}_{\boldsymbol{w}_t}, \boldsymbol{x} \rangle - \mu$ has a limit as $\lim_{t\to\infty} r_t = 0$. Nevertheless, this remains a challenging task even if it has $d = 2$. The update of GD over variables $\boldsymbol{w}_{t,\pm}$ involves complicated matrix-vector computations and adds to the difficulties of analyzing the iteration of $r_t$ and other important quantities. Therefore, the first step is to find a simple-to-tackle iteration that is equivalent to and completely determines the GD dynamics over the original model. This is entailed in Appendix A.1, in which by Lemma 4 and Lemma 5 we show that there exists an iteration of quadruplet $(a_t, a'_t, b'_t, b_t)$ that completely determines the trajectory of GD, and $r_t$ can be expressed as a linear combination of the quadruplet.

With $r_t$ expressed via simpler variables and updates, we continue to show that, first, $r_t$ converges to zero, and also, each variable in quadruplet can be properly bounded. To make the picture clearer, we divide the entire trajectory of the quadruplet into two or three phases depending the the value of $\mu\eta$. This corresponds to what we have discussed in the empirical observations, Section 3. The first case $\eta\mu \in (0, 1)$ results in two phases while the second case has $\eta\mu > 1$ and three phases.

We notice that, in both cases, there exists an initial phase. We show that in this phase $a'_t$ and $b'_t$ will decrease to zero in fast speed. This allows us to analyze a simpler iteration by regarding $a'_t$ and $b'_t$ as constant. The analysis of the initial phase is presented in Appendix A.2

The first case $\eta\mu \in (0, 1)$ has only two phases and its second phase is featured by the fact that $|r_t|$ always strictly contracts, and convergence can be easily established in a straightforward way. Nevertheless, in the second phase of case $\eta\mu > 1$, the iteration $r_t$ does not necessarily contract. On the opposite, the envelope of $r_t$ might increase during its oscillation, until it saturates. However, if condition $|x| \in (0, \frac{1}{\eta\mu})$ is satisfied, it can be shown that a *phase transition* always occurs such that $r_t$ begins to shrink. In this third phase, the convergence of $r_t$ can be proven using a similar technique for the case $\eta\mu \leq 1$. The proof of small step-size $\eta\mu \in (0, 1)$ is presented in Appendix A.3, and proof of $\eta\mu \in (1, \frac{3\sqrt{2}-2}{2})$ is in Appendix A.4. For even larger step-size and the proof of Proposition 1, please refer to Appendix A.5.

## A.1  LINEAR ALGEBRA AND SIMPLIFICATION OF GD DYNAMICS

In this subsection, we start to present the proof of the main theorem by finding a simple iteration that completely describes the behavior of GD. The major intermediate result of this subsection is summarized in Lemma 5 and Lemma 6. For readers who are not interested in linear algebra computation details, please skip the part and move directly to the aforementioned lemmas and the subsequent discussion.

The first part of this section focuses on deriving the equivalent iteration of quadruplet $(a_t, a'_t, b_t, b'_t)$ by analyzing the solution space of linear system $\langle \boldsymbol{x}, \boldsymbol{\beta} \rangle = y$. Let us define the following vectors $\boldsymbol{\beta}_0, \boldsymbol{\beta}_1 \in \mathbb{R}^2$:

$$\boldsymbol{\beta}_0 = (1, x), \qquad \text{and} \qquad \boldsymbol{\beta}_1 = (x, -1). \tag{4}$$

Notice that the collection $\{\boldsymbol{\beta} : i = 0, 1\}$ constitutes a linear independent and orthogonal basis in $\mathbb{R}^2$. Now consider the solution space of linear system $\langle \boldsymbol{x}, \boldsymbol{\beta} \rangle = y$, where $\boldsymbol{x} := (1, x) \in \mathbb{R}^{1\times2}$, $y = \langle \boldsymbol{x}, \boldsymbol{\beta}^* \rangle = \mu \in \mathbb{R}$ and $\boldsymbol{\beta} \in \mathbb{R}^2$. Then $\langle \boldsymbol{x}, \boldsymbol{\beta} \rangle = y$ is an underdetermined linear system with solution space as $\text{null}(\boldsymbol{x}) + \boldsymbol{\beta}^*$, where $\text{null}(\boldsymbol{x})$ is the null space of $\boldsymbol{x}$. Moreover, let $\text{null}^\perp(\boldsymbol{x})$ denote

the subspace orthogonal to $\text{null}(X)$. The dimensions of the null space and the complimentary space are, respectively,

$$\dim \text{null}(\boldsymbol{x}) = 2 - 1 = 1, \qquad \text{and} \qquad \dim \text{null}^{\perp}(\boldsymbol{x}) = 1.$$

It is easy to conclude that $\{\boldsymbol{\beta}_1\}$ and $\{\boldsymbol{\beta}_0\}$ spans $\text{null}(\boldsymbol{x})$ and $\text{null}^{\perp}(\boldsymbol{x})$, respectively.

To show the convergence of GD iteration in (2), it suffices to prove $\boldsymbol{\beta}_{\boldsymbol{w}_t} - \boldsymbol{\beta}^* \in \text{null}(\boldsymbol{x})$ when $t$ goes to infinity. To this end, we write $\boldsymbol{w}_{t,\pm}^2 \mp \frac{1}{2}\boldsymbol{\beta}^*$ as linear combination of $\{\boldsymbol{\beta}_0, \boldsymbol{\beta}_1\}$:

$$\boldsymbol{w}_{t,+}^2 = \sum_{i=0}^{1} p_t^i \boldsymbol{\beta}_i + \frac{1}{2}\boldsymbol{\beta}^*, \qquad \boldsymbol{w}_{t,-}^2 = \sum_{i=0}^{1} q_t^i \boldsymbol{\beta}_i - \frac{1}{2}\boldsymbol{\beta}^*. \tag{5}$$

This is equivalent to saying

$$\boldsymbol{\beta}_{\boldsymbol{w}_t} = \sum_{i=0}^{1} (p_t^i - q_t^i) \cdot \boldsymbol{\beta}_i + \boldsymbol{\beta}^*.$$

It is easy to derive that $r_t = (1 + x^2) \cdot (p_t^0 - q_t^0)$. We use compact notations $\boldsymbol{p}_t = (p_t^0, p_t^1)$ and $\boldsymbol{q}_t = (q_t^0, q_t^1)$, which allows to write the iteration using matrix-vector multiplications. In this way, the following lemma characterizes the evolution of $(\boldsymbol{p}_t, \boldsymbol{q}_t)$.

**Lemma 4.** *Consider the GD dynamics in (2) and the decomposition in Eq. (5). Then $(\boldsymbol{p}_t, \boldsymbol{q}_t)$ formalizes the following recurrences:*

$$\boldsymbol{p}_{t+1} = \left(\boldsymbol{I} - 2\eta r_t \boldsymbol{A} + \eta^2 r_t^2 \boldsymbol{B}\right) \cdot \boldsymbol{p}_t - \left(2\eta r_t - \eta^2 r_t^2\right) \cdot \frac{\mu \boldsymbol{\beta}^*}{2(1 + x^2)},$$

$$\boldsymbol{q}_{t+1} = \left(\boldsymbol{I} + 2\eta r_t \boldsymbol{A} + \eta^2 r_t^2 \boldsymbol{B}\right) \cdot \boldsymbol{q}_t - \left(2\eta r_t + \eta^2 r_t^2\right) \cdot \frac{\mu \boldsymbol{\beta}^*}{2(1 + x^2)} \tag{6}$$

*where $r_t = (1 + x^2) \cdot (p_t^0 - q_t^0)$, $\boldsymbol{A}, \boldsymbol{B} \in \mathbb{R}^{2 \times 2}$ are matrices defined as $\boldsymbol{A} = \frac{1}{1+x^2}\begin{pmatrix} 1 + x^3 & x(1-x) \\ x(1-x) & x(1+x) \end{pmatrix}$ and $\boldsymbol{B} = \frac{1}{1+x^2}\begin{pmatrix} 1 + x^4 & x(1-x^2) \\ x(1-x^2) & 2x^2 \end{pmatrix}$.*

*Proof.* The GD iteration in (2) indicates the following update of $\boldsymbol{w}_{t,\pm}$'s:

$$\boldsymbol{w}_{t+1,+} = \boldsymbol{w}_{t,+} - \eta r_t \boldsymbol{x} \odot \boldsymbol{w}_{t,+}, \qquad \boldsymbol{w}_{t+1,-} = \boldsymbol{w}_{t,-} + \eta r_t \boldsymbol{x} \odot \boldsymbol{w}_{t,-}.$$

We tackle $\boldsymbol{w}_{t,+}$ first and write down the expansion of $\boldsymbol{w}_{t,+}^2$ using elementwise products:

$$\boldsymbol{w}_{t+1,+}^2 = \left(\boldsymbol{1} - 2\eta r_t \boldsymbol{x} + \eta^2 r_t^2 \boldsymbol{x}^2\right) \odot \boldsymbol{w}_{t,+}^2$$
$$= \boldsymbol{w}_{t,+}^2 - 2\eta r_t \left(\boldsymbol{x} \odot \boldsymbol{w}_{t,+}^2\right) + \eta^2 r_t^2 \left(\boldsymbol{x}^2 \odot \boldsymbol{w}_{t,+}^2\right).$$

Since the linear independent set $\{\boldsymbol{\beta}_0, \boldsymbol{\beta}_1\}$ spans $\mathbb{R}^2$, we can represent $\boldsymbol{x} \odot \boldsymbol{w}_{t,+}^2$ as a linear combination of vectors from the basis:

$$\boldsymbol{x} \odot \boldsymbol{w}_{t,+}^2 = \sum_{i=0}^{1} \frac{\langle \boldsymbol{x} \odot \boldsymbol{w}_{t,+}^2, \boldsymbol{\beta}_i \rangle}{\|\boldsymbol{\beta}_i\|^2} \cdot \boldsymbol{\beta}_i$$

by plugging in Eq.(5)

$$= \sum_{i=0}^{1} \sum_{j=0}^{1} \frac{p_t^j \langle \boldsymbol{x} \odot \boldsymbol{\beta}_j, \boldsymbol{\beta}_i \rangle}{\|\boldsymbol{\beta}_i\|^2} \cdot \boldsymbol{\beta}_i + \frac{1}{2} \sum_{i=1}^{1} \frac{\langle \boldsymbol{x} \odot \boldsymbol{\beta}^*, \boldsymbol{\beta}_i \rangle}{\|\boldsymbol{\beta}_i\|^2} \cdot \boldsymbol{\beta}_i.$$

Similarly, we can expand $\boldsymbol{x}^2 \odot \boldsymbol{w}_{t,+}^2$ as a linear combination of $\boldsymbol{\beta}_0, \boldsymbol{\beta}_1$:

$$\boldsymbol{x}^2 \odot \boldsymbol{w}_{t,+}^2 = \sum_{i=0}^{1} \frac{\langle \boldsymbol{x}^2 \odot \boldsymbol{w}_{t,+}^2, \boldsymbol{\beta}_i \rangle}{\|\boldsymbol{\beta}_i\|^2} \cdot \boldsymbol{\beta}_i$$

$$= \sum_{i=0}^{1} \sum_{j=0}^{1} \frac{p_t^j \langle \boldsymbol{x}^2 \odot \boldsymbol{\beta}_j, \boldsymbol{\beta}_i \rangle}{\|\boldsymbol{\beta}_i\|^2} \cdot \boldsymbol{\beta}_i + \frac{1}{2} \sum_{i=1}^{1} \frac{\langle \boldsymbol{x}^2 \odot \boldsymbol{\beta}^*, \boldsymbol{\beta}_i \rangle}{\|\boldsymbol{\beta}_i\|^2} \cdot \boldsymbol{\beta}_i.$$

In the meanwhile, since $\boldsymbol{w}_{t+1,+}^2$ admits the following decomposition with coefficients $\boldsymbol{p}_{t+1}$:

$$\boldsymbol{w}_{t+1,+}^2 = \sum_{i=0}^{1} p_{t+1}^i \boldsymbol{\beta}_i + \frac{1}{2}\boldsymbol{\beta}^*, \tag{7}$$

we are able to compute by comparing the coefficients in Eq.(7) for $i = 0, 1$

$$p_{t+1}^i = p_t^i - 2\eta r_t \sum_{j=0}^{d} \frac{\langle \boldsymbol{x} \odot \boldsymbol{\beta}_j, \boldsymbol{\beta}_i \rangle}{\|\boldsymbol{\beta}_i\|^2} \cdot p_t^j + \eta^2 r_t^2 \sum_{j=0}^{d} \frac{\langle \boldsymbol{x}^2 \odot \boldsymbol{\beta}_j, \boldsymbol{\beta}_i \rangle}{\|\boldsymbol{\beta}_i\|^2} \cdot p_t^j$$

$$- 2\eta r_t \frac{\langle \boldsymbol{x} \odot \boldsymbol{\beta}^*, \boldsymbol{\beta}_i \rangle}{\|\boldsymbol{\beta}_i\|^2} + \eta^2 r_t^2 \frac{\langle \boldsymbol{x}^2 \odot \boldsymbol{\beta}^*, \boldsymbol{\beta}_i \rangle}{\|\boldsymbol{\beta}_i\|^2}.$$

This can be expressed with compact notation via matrix-vector multiplication as

$$\boldsymbol{p}_{t+1} = \left(\boldsymbol{I} - 2\eta r_t \boldsymbol{A} + \eta^2 r_t^2 \boldsymbol{B}\right) \cdot \boldsymbol{p}_t - 2\eta r_t \frac{\boldsymbol{\mu}}{2} + \eta^2 r_t^2 \frac{\boldsymbol{\nu}}{2},$$

where $(\boldsymbol{A})_{ij} = A_{ij} = \frac{\langle \boldsymbol{x} \odot \boldsymbol{\beta}_j, \boldsymbol{\beta}_i \rangle}{\|\boldsymbol{\beta}_i\|^2}$, $(\boldsymbol{B})_{ij} = B_{ij} = \frac{\langle \boldsymbol{x}^2 \odot \boldsymbol{\beta}_j, \boldsymbol{\beta}_i \rangle}{\|\boldsymbol{\beta}_i\|^2}$, $(\boldsymbol{\mu})_i = \mu_i = \frac{\langle \boldsymbol{x} \odot \boldsymbol{\beta}^*, \boldsymbol{\beta}_i \rangle}{\|\boldsymbol{\beta}_i\|^2}$, and $(\boldsymbol{\nu})_i = \nu_i = \frac{\langle \boldsymbol{x}^2 \odot \boldsymbol{\beta}^*, \boldsymbol{\beta}_i \rangle}{\|\boldsymbol{\beta}_i\|^2}$ for $i, j \in [d]$. Noticing the symmetry between $\boldsymbol{p}_t$ and $\boldsymbol{q}_t$, we obtain the obtain the update of $\boldsymbol{q}_t$:

$$\boldsymbol{q}_{t+1} = \left(\boldsymbol{I} + 2\eta r_t \boldsymbol{A} + \eta^2 r_t^2 \boldsymbol{B}\right) \cdot \boldsymbol{q}_t - 2\eta r_t \frac{\boldsymbol{\mu}}{2} - \eta^2 r_t^2 \frac{\boldsymbol{\nu}}{2}.$$

It still remains to determine the exact value of $A_{ij}$'s, $B_{ij}$'s and $\mu_i$'s. We calculate matrices $\boldsymbol{A}$ and $\boldsymbol{B}$ by discussing the following cases:

1. **Case** $i = 0$, $j = 1$: $A_{ij} = \frac{x(1-x)}{1+x^2}$, $B_{ij} = \frac{x(1-x^2)}{1+x^2}$;

2. **Case** $i = 0$, $j = 0$: $A_{ij} = \frac{1+x^3}{1+x^2}$, $B_{ij} = \frac{1+x^3}{1+x^2}$;

3. **Case** $i = 1$, $j = 1$: $A_{ij} = \frac{x(x+1)}{1+x^2}$, $B_{ij} = \frac{2x^2}{1+x^2}$;

4. **Case** $i = 1$, $j = 0$: $A_{ij} = \frac{x(1-x)}{1+x^2}$, $B_{ij} = \frac{x(1-x^2)}{1+x^2}$.

Similarly, $\boldsymbol{\mu}$ and $\boldsymbol{\nu}$ are computed as

1. **Case** $i = 0$: $\mu_i = \nu_i = \frac{\mu}{1+x^2}$;

2. **Case** $i = 1$: $\mu_i = \nu_i = \frac{\mu x}{1+x^2}$.

Noticing that $\boldsymbol{\mu} = \boldsymbol{\nu} = \mu\boldsymbol{\beta}_0/(1+x^2)$, we finish the proof. $\qquad\square$

We do not stop at the iterations of $(\boldsymbol{p}_t, \boldsymbol{q}_t)$ because it is still hard to analyze their updates via matrix-vector multiplication. It requires further simplification. Let us define vectors

$$\boldsymbol{v}_1 = (1, x), \qquad \boldsymbol{v}_2 = (x, -1).$$

We write down the (non-standard) eigencomposition of matrices $\boldsymbol{A}$ and $\boldsymbol{B}$ define in Lemma 4:

$$\boldsymbol{A} = \lambda_1(\boldsymbol{A})\boldsymbol{v}_1\boldsymbol{v}_1^\top + \lambda_2(\boldsymbol{A})\boldsymbol{v}_2\boldsymbol{v}_2^\top$$

$$\boldsymbol{B} = \lambda_1(\boldsymbol{B})\boldsymbol{v}_1\boldsymbol{v}_1^\top + \lambda_2(\boldsymbol{B})\boldsymbol{v}_2\boldsymbol{v}_2^\top,$$

where we have

$$\lambda_1(\boldsymbol{A}) = \lambda_1(\boldsymbol{B}) = 1, \qquad \lambda_2(\boldsymbol{A}) = x, \qquad \text{and} \qquad \lambda_2(\boldsymbol{B}) = x^2.$$

This suggests that $\boldsymbol{A}$ and $\boldsymbol{B}$ share the same eigenspace. Moreover, $\boldsymbol{\beta}_0$ can be written as $\boldsymbol{\beta}_0 = \boldsymbol{v}_1$. As a result, we write $\boldsymbol{p}_t, \boldsymbol{q}_t \in \mathbb{R}^2$ as the linear combination of $\boldsymbol{v}_1, \boldsymbol{v}_2$:

$$\boldsymbol{p}_t = \left(a_t - \frac{\mu}{2(1+x^2)}\right)\boldsymbol{v}_1 + b_t\boldsymbol{v}_2, \qquad \boldsymbol{q}_t = \left(a_t' + \frac{\mu}{2(1+x^2)}\right)\boldsymbol{v}_1 + b_t'\boldsymbol{v}_2. \tag{8}$$

The iteration of $(a_t, b_t, a_t', b_t')$ is characterized by the following lemma.

**Lemma 5.** *Consider the iteration of $(\boldsymbol{p}_t, \boldsymbol{q}_t)$ defined in (6) and the decomposition in Eq. (8). Then $(a_t, b_t, a'_t, b'_t)$ formalizes the following recurrences*

$$\begin{cases} a_{t+1} = \left(1 - \eta r_t\right)^2 \cdot a_t \\ a'_{t+1} = \left(1 + \eta r_t\right)^2 \cdot a'_t \\ b_{t+1} = \left(1 - x \cdot \eta r_t\right)^2 \cdot b_t \\ b'_{t+1} = \left(1 + x \cdot \eta r_t\right)^2 \cdot b'_t, \end{cases}$$

*where we have*

$$r_t = (1 + x^2) \cdot \left((a_t - a'_t) + x \cdot (b_t - b'_t) - \frac{\mu}{1 + x^2}\right).$$

*Additionally, the initialization satisfies $a_0 = a'_0 = b_0 = b'_0 = \frac{\alpha}{1 + x^2}$.*

*Proof.* We consider the decomposition of $\boldsymbol{p}_t$ as in Eq. (8) and calculate the expansion of $\boldsymbol{p}_{t+1}$ using Lemma 4:

$$\boldsymbol{p}_{t+1} = \left(\boldsymbol{I} - 2\eta r_t \boldsymbol{A} + \eta^2 r_t^2 \boldsymbol{B}\right) \cdot \left(a_t \boldsymbol{v}_1 + b_t \boldsymbol{v}_2 - \frac{\mu \boldsymbol{v}_1}{2(1 + x^2)}\right) - (2\eta r_t - \eta^2 r_t^2) \cdot \frac{\mu \boldsymbol{v}_1}{2(1 + x^2)},$$

since $\lambda_i(\boldsymbol{A}), \lambda_i(\boldsymbol{B})$ are the corresponding (non-standard) eigenvalues of $\boldsymbol{A}$ and $\boldsymbol{B}$ for $i = \{1, 2\}$,

$$= \left(1 - 2\eta r_t \lambda_1(\boldsymbol{A}) + \eta^2 r_t^2 \lambda_1(\boldsymbol{B})\right) \cdot a_t \boldsymbol{v}_1 - (1 - 2\eta r_t + \eta^2 r_t^2) \cdot \frac{\mu \boldsymbol{v}_1}{2(1 + x^2)}$$

$$+ \left(1 - 2\eta r_t \lambda_2(\boldsymbol{A}) + \eta^2 r_t^2 \lambda_2(\boldsymbol{B})\right) \cdot b_t \boldsymbol{v}_2 - (2\eta r_t - \eta^2 r_t^2) \cdot \frac{\mu \boldsymbol{v}_1}{2(1 + x^2)}$$

$$= (1 - \eta r_t)^2 \cdot a_t + (1 - x \cdot \eta r_t)^2 \cdot b_t - \frac{\mu \boldsymbol{v}_1}{2(1 + x^2)}.$$

By comparing the expansion with the following identity

$$\boldsymbol{p}_{t+1} = a_{t+1} \boldsymbol{v}_1 + b_{t+1} \boldsymbol{v}_2 - \frac{\mu \boldsymbol{v}_1}{2(1 + x^2)},$$

we obtain the following updates of coefficients

$$a_{t+1} = \left(1 - \eta r_t\right)^2 \cdot a_t, \qquad b_{t+1} = \left(1 - x \cdot \eta r_t\right)^2 \cdot b_t.$$

Noticing the symmetry between $\boldsymbol{p}_t$ and $\boldsymbol{q}_t$ and their update, we also obtain the iteration of $a'_t$ and $b'_t$ as

$$a'_{t+1} = \left(1 + \eta r_t\right)^2 \cdot a'_t, \qquad b'_{t+1} = \left(1 + x \cdot \eta r_t\right)^2 \cdot b'_t.$$

Finally, simple calculation shows $r_t = (1 + x^2) \cdot \left((a_t - a'_t) + x \cdot (b_t - b'_t)\right) - \mu.$ $\qquad \square$

Since our major goal is to prove the convergence of $r_t$, we want to express the update of $r_t$ directly. This is entailed in the next lemma.

**Lemma 6.** *The update of $r_t$ follows the iteration below:*

$$r_{t+1} = \left(1 - 2\eta\left(\mu + r_t - c_x \cdot (b_t - b'_t)\right)\right) \cdot r_t + \eta^2 r_t^2 \cdot \left(\mu + r_t - c'_x \cdot (b_t - b'_t)\right)$$

$$- (1 + x^2) \cdot 4\eta r_t \cdot \left(a'_t + x^2 \cdot b'_t\right).$$

*where $c_x = x(1 - x)(1 + x^2)$ and $c'_x = x(1 - x^2)(1 + x^2)$.*

*Proof.* We expand the update of $r_t$ as

$$r_{t+1} = (1 + x^2) \cdot \left((a_{t+1} - a'_{t+1}) + x \cdot (b_{t+1} - b'_{t+1}) - \frac{\mu}{1 + x^2}\right)$$

by plugging the iteration in Lemma 5,

$$= (1+x^2) \cdot \bigg( (1-\eta r_t)^2 \cdot a_{t+1} - (1+\eta r_t)^2 \cdot a'_{t+1} + x \cdot (1-x\eta r_t)^2 \cdot b_{t+1}$$

$$- x \cdot (1-x\eta r_t)^2 \cdot b'_{t+1} - \frac{\mu}{1+x^2} \bigg)$$

by merging terms in order of $r_t$'s,

$$= (1+x^2) \cdot \bigg( (a_t - a'_t) + x \cdot (b_t - b'_t) - \frac{\mu}{1+x^2} \bigg) - 2\eta r_t \cdot (1+x^2) \cdot \bigg( (a_t + a'_t) + x^2 \cdot (b_t + b'_t) \bigg)$$

$$+ \eta^2 r_t^2 \cdot (1+x^2) \cdot \bigg( (a_t - a'_t) + x^3 \cdot (b_t - b'_t) \bigg)$$

$$= (1+x^2) \cdot \bigg( (a_t - a'_t) + x \cdot (b_t - b'_t) - \frac{\mu}{1+x^2} \bigg)$$

$$- 2\eta r_t \cdot (1+x^2) \cdot \bigg( (a_t - a'_t) + x \cdot (b_t - b'_t) - \frac{\mu}{1+x^2} \bigg)$$

$$+ 2\eta r_t \cdot (1+x^2) \cdot \bigg( -2a'_t - 2x^2 \cdot b'_t + x(1-x) \cdot (b_t - b'_t) - \frac{\mu}{1+x^2} \bigg)$$

$$+ \eta^2 r_t^2 \cdot (1+x^2) \cdot \bigg( (a_t - a'_t) + x \cdot (b_t - b'_t) - \frac{\mu}{1+x^2} \bigg)$$

$$- \eta^2 r_t^2 \cdot (1+x^2) \cdot \bigg( x(1-x^2) \cdot (b_t - b'_t) - \frac{\mu}{1+x^2} \bigg).$$

By noticing the equality of $r_t$ as in Lemma 5, we obtain

$$r_{t+1} = \bigg( 1 - 2\eta\big(\mu + r_t - c_x \cdot (b_t - b'_t)\big) \bigg) \cdot r_t + \eta^2 r_t^2 \cdot \bigg( \mu + r_t - c'_x \cdot (b_t - b'_t) \bigg)$$

$$- (1+x^2) \cdot 4\eta r_t \cdot \big( a'_t + x^2 \cdot b'_t \big)$$

where $c_x = x(1+x^2)(1-x)$ and $c'_x = x(1+x^2)(1-x^2)$. $\qquad\square$

For future convenience, we introduce below several compact notations. First, denote $\Delta a_t = a_t - a'_t$ and $\Delta b_t = b_t - b'_t$. This allows us to define $\alpha_t$, $\beta_t$ and $\gamma_t$

$$\alpha_t = 2 - 2\eta\big(\mu - c_x \Delta b_t\big), \quad \beta_t = 2\eta - \eta^2\big(\mu + r_t - c'_x \Delta b_t\big), \quad \gamma_t = (1+x^2) \cdot 4\eta\big(a'_t + x^2 \cdot b'_t\big).$$

The above notations allow us to rewrite the update of $r_t$ in Lemma 6 in a more compact form:

$$r_{t+1} = -(1 - \alpha_t + \gamma_t) \cdot r_t - \beta_t \cdot r_t^2. \tag{9}$$

## A.2 Initial phase and Transition to Oscillation

In Appendix A.1, we have shown that the GD iteration can be equivalently described by the iteration of quadruplet $(a_t, a'_t, b_t, b'_t)$ in Lemma 5. Moreover, we express $r_{t+1}$ as a function of $r_t$ and the quadruplet in Lemma 6. In this subsection, we will analyze the behavior of $(a_t, a'_t, b_t, b'_t)$ and $r_t$ in the initial phases. It can be easily derived from that the initialization $\mathbf{w}_{0,\pm} = \alpha\mathbf{1}$ corresponds to the following initial value of the quadruplet

$$a_0 = a'_0 = b_0 = b'_0 = \frac{\mu}{2(1+x^2)}.$$

As a result, we have $r_0 = -\mu$.

From the iteration in Lemma 5, it is obvious that all of $a_t, a'_t, b_t,$ and $b'_t$ are non-negative for any $t$. We will use this simple property without reference to it. When $\alpha$ is set to be sufficiently small, the value of each term in the quadruplet will be negligible compared with $|r_t|$ when $t$ is small. As a result, $r_t$ remains negative and stable as $r_t \approx -\Theta(\mu)$ holds in the initial phase, which leads

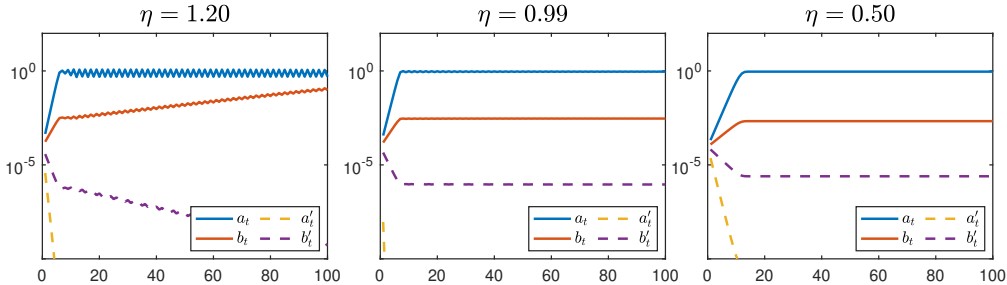

Figure 9: Stable behavior of $a'_t$ and $b'_t$ iterations in Lemma 5. Under initialization $a_0 = a'_0 = b_0 = b'_0 = \frac{\mu}{2(1+x^2)}$, both $a'_t$ and $b'_t$ will decrease to zero in initial several iterations. We use different step-sizes in each plot, which corresponds to EoS with $\eta\mu > 1$, with $\eta\mu \leq 1$ and GF regime, respectively. We fix other parameters as $w = 0.3$, $\mu = 1$ and $\alpha = 0.01$.

to the fast increase of $a_t$, $b_t$ and also the fast decrease of $a'_t, b'_t$ to **zero** in a few iterations. This phenomenon is empirically illustrated in Figure 9. The observation leads to two important results. First, the fast decrease of $a'_t$ and $b'_t$ allows us to further simplify the dynamics in Lemma 5 by regarding $a'_t \equiv b'_t \equiv 0$ for any $t$ larger than some constant $t_0$, where $t_0$ marks the end of the initial phase. In the meanwhile, since $r_t < 0$ holds in the initial phase, we have

$$\frac{a_{t+1}}{b_{t+1}} = \frac{(1 - \eta r_t)^2}{(1 - x \cdot \eta r_t)^2} \cdot \frac{a_t}{b_t} \approx e^{-\Theta(\eta r_t)} \cdot \frac{a_t}{b_t} \approx e^{\Theta(\eta\mu)} \cdot \frac{a_t}{b_t},$$

which suggests the increase of $a_t$'s outrun $b_t$'s. As a result, the update of $r_t$ is approximate as

$$r_{t+1} \approx (1 + x^2) \cdot a_{t+1} - \mu \approx (1 - \eta r_t)^2 (r_t + \mu) - \mu$$
$$= -(2\mu\eta - 1)r_t - (2\eta - \eta^2\mu)r_t^2 + \eta^2 r_t^3,$$

which generates an increasing sequence (therefore decreasing in absolute value) when $r_t < -\Theta(\mu)$.

It can be possible that the value of $r_t$ decreases monotonically for $t$ after $t_0$ and hence the trajectory falls into the GF regime. On the contrary, in the EoS regime, while $r_t$ is decreased enough, $a_t$ can increase to reach the level and surpass $\Theta(|r_t|)$, which causes an oscillation to start and hence ends the initial phase. The qualitative analysis is more formally analyzed in the following lemma.

**Lemma 7.** *Suppose that $\mu\eta \in (0, 2)$ and $\alpha \ll O(1)$. There exist some $\tau = \Omega_{\eta,\mu}\left(\log\left(\frac{1}{\alpha^2}\right)\right)$ such that for any $t < \tau$, $r_t < -\frac{\mu}{2}$ is true. As a result, the following facts are true:*

$$b_\tau \leq \mu^{-\Theta(1)} \cdot \alpha^{2-C}, \qquad a'_\tau \leq \Theta(\frac{\alpha^4}{\mu}), \qquad b'_\tau \leq \mu^{\Theta(1)} \cdot \alpha^{2+C}.$$

*where $C$ is some constant between $(0, 2)$.*

*Proof.* Let $\tau$ be the first $t$ such that $r_t < -\frac{\mu}{2}$ does not hold. Suppose such a $\tau$ does not exist and $r_t < -\frac{\mu}{2}$ holds for any integer $t$. We observe the following inequality is true

$$a'_{t+1} = (1 + \eta r_t)^2 \cdot a'_t < a'_t.$$

Repeating the steps results in $a'_t < a'_0$. Due to the same argument, it holds that $b'_t < b'_0 = b_0 < b_t$. We consider the lower bound of $a_t$: since $r_t < -\frac{\mu}{2}$ is true for any $t$ by assumption,

$$a_t \geq \left(1 + \frac{\eta\mu}{2}\right)^2 \cdot a_{t-1} \geq \cdots \geq \left(1 + \frac{\eta\mu}{2}\right)^{2t} \cdot \alpha_0 = \left(1 + \frac{\eta\mu}{2}\right)^{2t} \cdot \frac{\alpha^2}{2(1 + x^2)}.$$

We continue and use in the identity of $r_t$ in Lemma 5:

$$-\frac{\mu}{2} > r_t = (1 + x^2)\Big((a_t - a'_t) + x \cdot (b_t - b'_t)\Big) - \mu$$

$$\geq \left(1 + \frac{\eta\mu}{2}\right)^{2t} \cdot \frac{\alpha^2}{2} - \mu - \frac{\alpha^2}{2}$$

$$\geq \frac{\alpha^2}{4} \cdot \left(1 + \frac{\eta\mu}{2}\right)^{2t} - \mu$$

Rearranging yields $\frac{2\mu}{\alpha^2} \geq \left(1 + \frac{\eta\mu}{2}\right)^{2t}$, which should holds for $t$ by assumption. This is impossible because $\alpha$, $\eta$ and $\mu$ are constants, and we conclude by contradiction that $\tau$ exists and $\tau = O\left(\frac{1}{\mu\eta} \log\left(\frac{\mu}{\alpha^2}\right)\right)$.

We proceed to the lower bound of $\tau$. We first notice the following is true for any $t < \tau$:

$$r_{t+1} = (1 + x^2) \cdot \left((a_{t+1} - a'_{t+1}) + x \cdot (b_{t+1} - b'_{t+1})\right) - \mu \geq -\mu.$$

This is because $r_t < 0$ holds for any $t < \tau$ and therefore the above conclusion $a'_{t+1} \leq a_{t+1}$, $b'_{t+1} \leq b_{t+1}$ is still true. This allows to lower bound $a_\tau$ as

$$a_\tau \leq e^{-2\eta r_{\tau-1}} \cdot a_{\tau-1} \leq e^{2\mu\eta\tau} \cdot a_0.$$

In the meanwhile, since $\tau$ is the first $t$ such that $r_t > -\frac{\mu}{2}$ no longer holds, we have the following inequality

$$a_\tau = \frac{r_\tau + \mu}{1 + x^2} - x \cdot (b_\tau - b'_\tau) - a'_\tau \geq \frac{-\mu/2 + \mu}{1 + x^2} - x \cdot a_\tau - O(\alpha^2),$$

by rearranging the equality of $r_t$. We notice $x^2 < 1$ and therefore $a_\tau \geq \frac{\mu}{4(1+x^2)} \geq \mu/8$. Combining the above result, we obtain $a_0 \cdot e^{2\mu\eta\tau} \geq \mu/8$, which implies $\tau \geq \Omega\left(\frac{1}{\mu\eta} \log\left(\frac{\mu}{\alpha^2}\right)\right)$. It remains to bound for $b_t$, $a'_t$ and $b'_t$. For $a'_t$, it holds that

$$a'_\tau a_\tau \leq (1 - \eta r_{\tau-1})^2 \cdot (1 - \eta r_{\tau-1})^2 \cdot a'_\tau a_\tau < a'_{\tau-1} a_{\tau-1} < a'_0 a_0 = \Theta(\alpha^4),$$

which suggests $a'_\tau \leq \Theta(\mu/\alpha^4)$. For $b_t$, it can be computed as

$$b_\tau \leq e^{-2x\eta r_{\tau-1}} \cdot b_{\tau-1} \leq e^{2x\eta\tau} \cdot b_0 \leq \Theta(\alpha^2) \cdot e^{\Theta(x) \cdot \log(\mu/\alpha^2)} \leq \mu^{-\Theta(1)} \cdot \alpha^{2-C}$$

where $C$ is some constant between $(0, 2)$. Combining with

$$b'_\tau b_\tau \leq b'_0 b_0 \leq \Theta(\alpha^4),$$

it holds that $b'_\tau \leq \mu^{\Theta(1)} \cdot \alpha^{2+C} \ll b\tau$. $\qquad\square$

In the statement of Theorem 1 and Theorem 2, we assume that change of sign starts after the initial phase as $r_t r_{t+1}$ holds for any $t \geq t_0$. Lemma 7 suggests a lower bound of $t_0 \geq \tau = \Omega_{\eta,\mu}(\log(1/\alpha^2))$ as oscillation does not start when $r_t < -\mu/2$ still holds. Moreover, when $\alpha$ is sufficiently small, Lemma 7 indicates that $a'_t$ and $b'_t$ will decrease very quickly to zero before the change of sign, and hence allows us to simplify the iteration in Lemma 5 by regarding

$$a'_t \equiv b'_t \equiv 0, \qquad \text{for} \qquad \forall t \geq t_0.$$

We use this approximation for any $t \geq t_0$. Now, we can write the iteration of $r_t$ as

$$r_{t+1} = -(1 - \alpha_t + \beta_t r_t) \cdot r_t \tag{10}$$

where $\alpha_t = 2 - 2\eta\big(\mu - c_x b_t\big)$ and $\beta_t = 2 - 2\eta\big(\mu + r_t - c'_x b_t\big)$. Under this setting, we establish the convergence of $r_t$ for both setting $\eta\mu \leq 1$ and $\eta\mu > 1$, as in the next two theorems, respectively.

### A.3 PROOF OF THEOREM 1: $\eta\mu < 1$

This subsection contains the convergence proof under the EoS regime with $\mu\eta < 1$. We define some notations before proceeding to its proof. Since our major focus is the convergence of $|r_t|$, it is more convenient to render the update of $(a_t, b_t)$ in 10 as an equivalent update of bivariate $(r_t, s_t)$ defined as following. We first express $a_t$ as a linear combination of $r_t$ and $b_t$

$$r_t = (1 + x^2) \cdot (a_t + x \cdot b_t) - \mu \qquad \Longrightarrow \qquad a_t = \frac{r_t + \mu}{1 + x^2} - x \cdot b_t.$$

Moreover, for notational convenience, we replace $b_t$ with $s_t$ where the latter is rescaled version of $b_t$ by a constant factor, i.e. $s_t = \eta c_x \cdot b_t$. To obtain the update of the new sequence $(r_t, s_t)$, we define the following polynomials of pair $(r, s)$ with fixed constants $\eta$, $\mu$ and $x$:

$$g_{\eta,\mu,x}(r, s) = -(2\mu\eta - 1)r - (2\eta - \mu\eta^2)r^2 + \eta^2 r^3 + 2rs - (1 + x)\eta r^2 s,$$

$$h_{\eta,\mu,x}(r, s) = s - 2x\eta rs + x^2\eta^2 r^2 s.$$

The subscript is omitted, i.e. $g(r, s) = g_{\eta,\mu,x}(r, s)$ when it causes no confusion. Comparing $g, h$ and the iteration of $(a_t, b_t)$ in Lemma 5 we obtain the following identities

$$r_{t+1} = g(r_t, s_t), \qquad s_{t+1} = h(r_t, s_t). \tag{11}$$

Therefore, we could employ the toolbox of dynamical systems to characterize the behavior of $(r_t, s_t)$. Besides, it is worth noticing the following identity holds:

$$\alpha_t = 2 - 2\eta\mu + 2\eta c_x b_t = 2 - 2\eta\mu + 2s_t,$$
$$\beta_t = 2\eta - \eta(\mu\eta + r_t\eta + (1 + x) \cdot s_t)$$

We will frequently use these equalities in the latter part.

**Proof outline.** By the end of the initial phase, the absolute value of negative $r_t$ has decreased sufficiently, and the oscillation $r_t r_{t+1} < 0$ starts. From our empirical observation in Section 3, the major characteristic of EoS regime with $\eta\mu < 1$ is that the envelope of $r_t$ enters the convergence phase immediately as the initial phase ends. This is because during the convergence phase, for any $r_t < 0$, it can be shown that

$$0 > r_{t+2} > (1 - \alpha_t)(1 - \alpha_{t+1}) \cdot r_t.$$

When the step-size is set to be smaller than $1/\mu$, $\alpha_t \geq 2 - 2\eta\mu > 0$ will hold for any $t$, which further implies the shrinkage of $|r_t|$'s.

We start the proof by noticing the simple results for $a_t$ and $b_t$'s.

**Lemma 8.** *For any $t$, if $r_t \in (-\mu, 0]$ is true, then the following inequalities hold:*

$$a_{t+1} > a_t, \quad b_{t+1} > b_t.$$

*Proof.* Let $r_t \in [-\mu, 0)$, then it holds that $1 - \eta r_t > 1$ and $1 - x \cdot \eta r_t > 1$. We obtain the following inequalities immediately from the update in Lemma 5:

$$a_{t+1} = (1 - \eta r_t)^2 \cdot a_t > a_t, \qquad b_{t+1} = (1 - x \cdot \eta r_t)^2 \cdot b_t \cdot b_t' > b_t,$$

and finish the proof. $\qquad\square$

**Lemma 9.** *For any $r_t < 0$ and $r_{t+1} > 0$, it holds that $r_{t+2} \geq (1 - \alpha_{t+1})(1 - \alpha_t) \cdot r_t$.*

*Proof.* Let $r_t < 0$ and $r_{t+1} > 0$. To obtain the lower bound, we first write down the expansion of $r_{t+2}$ using Eq.(10):

$$r_{t+2} = -(1 - \alpha_{t+1}) \cdot r_{t+1} - \beta_{t+1} \cdot r_{t+1}^2$$
$$= (1 - \alpha_{t+1})(1 - \alpha_t) \cdot r_t + (1 - \alpha_{t+1}) \cdot \beta_t \cdot r_t^2 - \beta_{t+1} \cdot r_{t+1}^2$$
$$= (1 - \alpha_{t+1})(1 - \alpha_t) \cdot r_t + \beta_t \cdot (r_t^2 - r_{t+1}^2) - \alpha_{t+1}\beta_t \cdot r_t^2 + (\beta_t - \beta_{t+1}) \cdot r_{t+1}^2 \tag{12}$$

By observing Eq. (10) again, we have

$$r_{t+1} + r_t = \alpha_t r_t - \beta_t r_t^2,$$

which allows the following decomposition of $r_t^2 - r_{t+1}^2$

$$r_t^2 - r_{t+1}^2 = (r_t - r_{t+1}) \cdot (r_t + r_{t+1}) = (r_t - r_{t+1}) \cdot (\alpha_t r_t - \beta_t r_t^2).$$

We plug the identity back to Eq.(12) and obtain

$$r_{t+2} = (1 - \alpha_{t+1})(1 - \alpha_t) \cdot r_t + \beta_t \cdot (r_t - r_{t+1}) \cdot (\alpha_t r_t - \beta_t r_t^2) - \alpha_{t+1}\beta_t' \cdot r_t^2 + (\beta_t - \beta_{t+1}) \cdot r_{t+1}^2$$
$$= (1 - \alpha_{t+1})(1 - \alpha_t) \cdot r_t + \alpha_t\beta_t \cdot r_t(r_t - r_{t+1}) - \beta_t^2 \cdot r_t^2(r_t - r_{t+1})$$
$$\qquad - \alpha_t\beta_t \cdot r_t^2 - (\alpha_t - \alpha_{t+1})\beta_t \cdot r_t^2 + (\beta_t - \beta_{t+1}) \cdot r_{t+1}^2$$
$$= (1 - \alpha_{t+1})(1 - \alpha_t) \cdot r_t - \alpha_t\beta_t \cdot r_t r_{t+1} - \beta_t^2 \cdot r_t^2(r_t - r_{t+1})$$
$$\qquad - (\alpha_t - \alpha_{t+1})\beta_t \cdot r_t^2 + (\beta_t - \beta_{t+1}) \cdot r_{t+1}^2.$$

We notice the following facts:

$$r_t - r_{t+1} < 0, \qquad \alpha_t - \alpha_{t+1} < 0, \qquad \beta_t - \beta_{t+1} > 0.$$

The first inequality is true due to $r_t < 0$ and $r_{t+1} > 0$. For the second one, it is easy to verify

$$\alpha_t - \alpha_{t+1} = 2\eta c_x \cdot \left(b_t - b_{t+1}\right) < 0$$

due to Lemma 8. For the last inequality, we expand using the definition of $\beta_t$

$$\begin{aligned}
\beta_t - \beta_{t+1} &= \eta^2 c_x' \cdot \left(b_t - b_{t+1}\right) - \eta^2 \cdot \left(r_t - r_{t+1}\right) \\
&= -\eta^2 (1 + x^2) \cdot \left((a_t - a_{t+1}) + x^3 \cdot (b_t - b_{t+1})\right) \\
&> 0.
\end{aligned}$$

where the second equality comes from $r_t = (1 + x^2) \cdot \left(a_t + x \cdot b_t\right) - \mu$, and the inequality is due to Lemma 8 again. Combining the above facts and the expansion of $r_{t+2}$ in Eq. (12), we attain the following inequality

$$r_{t+2} \geq (1 - \alpha_{t+1})(1 - \alpha_t) \cdot r_t.$$

$\square$

With the help of the above two lemmas, we are able to show the contraction of $r_t$'s envelope when $t \geq t_0$.

**Lemma 10.** *Suppose that $\eta\mu < 1$ and $r_t r_{t+1} < 0$ holds for any $t \geq t_0$. The iteration of $(r_t, s_t)$ converges to $(0, s_\infty)$ in a linear rate as*

$$|r_t| \leq \exp\left(-\Theta(\mu\eta) \cdot (t - t_0)\right) \cdot |r_{t_0}|.$$

*Moreover, it holds that $\lim_{t \to \infty} s_t \leq C \cdot \alpha^{\Theta(1)}$.*

*Proof.* Let us assume $r_t r_{t+1}$ holds for any $t \geq t_0$. To show the convergence of $r_t$, we only need to consider bounding $|r_t|$ when $r_t$ is negative, because for any $r_{t+1} > 0$, we have directly $|r_{t+1}| \leq O(|r_t|)$.

Now consider any $t \geq t_0$ with $r_t < 0$. We invoke Lemma 9 to obtain the lower bound for $r_{t+2}$, which is also negative by our assumption:

$$r_{t+2} \geq (1 - \alpha_t)(1 - \alpha_{t+1}) \cdot r_t.$$

To proceed, it requires to bound $\alpha_t$ and $\alpha_{t+1}$. We first show $1 - \alpha_t > 0$ and $1 - \alpha_{t+1} > 0$ holds by discussing two cases. In the first case, if $\beta_t \geq 0$, it holds that

$$0 < -\frac{r_{t+1}}{r_t} = 1 - \alpha_t + \beta_t r_t < 1 - \alpha_t$$

because $r_t < 0$. Then due to Eq. (10 and fact $r_{t+2}, r_t < 0$ we must also have $1 - \alpha_{t+1} > 0$. In the second case, suppose $\beta_t < 0$. By above argument we attain $\beta_{t+1} < \beta_t < 0$. Similarly, because $r_{t+1} r_{t+2} < 0$ and $r_{t+1} > 0$, we attain

$$0 < -\frac{r_{t+2}}{r_{t+1}} = 1 - \alpha_{t+1} + \beta_{t+1} r_{t+1} < 1 - \alpha_{t+1}.$$

As a result, $1 - \alpha_t > 0$ is also true. We also need to lower bound $\alpha_t$ and $\alpha_{t+1}$. Since $\mu\eta < 1$, by their definition and Lemma 8 we have immediately

$$\alpha_{t+1} \geq \alpha_t \geq 2(1 - \mu\eta) > 0.$$

As a result of the above discussion, we obtain $|r_{t+2}| \leq \left(1 - (2\mu\eta - 1)\right)^2 \cdot |r_t|$ because $2\mu\eta - 1 < 1$ for any $\mu\eta \in (0, 1)$. This suggests

$$|r_t| \leq \exp\left(\Theta(2\mu\eta - 1) \cdot (t - t_0)\right) \cdot |r_{t_0}|.$$

It remains to bound $s_t$'s. To this end, we focus on the iteration $t$ with $r_t > 0$ instead of $r_t < 0$, because from Lemma 8 $s_t > s_{t-1}$ if $r_t > 0$. For any such $t$, we compute

$$\begin{aligned}
\frac{s_{t+2}}{s_t} = (1 - x \cdot \eta r_{t+1})^2 \cdot (1 - x \cdot \eta r_t)^2 &\leq \exp\left(-x\eta \cdot (r_t + r_{t+1})\right) \\
&= \exp\left(-x\eta \cdot (r_t + r_{t+1})\right) \\
&= \exp\left(-x\eta\alpha_t r_t + x\eta\beta_t r_t^2\right) \\
&\leq \exp\left(2x\eta^2 r_t^2\right)
\end{aligned}$$

where the second equality is from $r_t + r_{t+1} = -\alpha_t r_t + \beta_t r_t^2$, an equivalent form of Eq. (10), and the last inequality come from $\alpha_t > 0$ and $r_t > 0$ and

$$\beta_t = 2\eta - \eta^2(\mu + r_t - c_x' b_t) = 2\eta - \eta^2(1 + x^2) \cdot (a_t + x^3 \cdot b_t) < 2\eta.$$

Repeating the above step, we obtain for any $t$ with $r_t > 0$

$$s_t = \exp\left(2x\eta^2 \cdot \sum_{\substack{i=t_0 \\ i \text{ even}}}^{t-2} r_t^2\right) \cdot s_{t_0} \leq \exp\left(2x\eta^2 r_{t_0}^2 \cdot \sum_{\substack{i=t_0 \\ i \text{ even}}}^{t-2} e^{-\Theta(2\mu\eta-1)}\right) \cdot s_{t_0}$$

$$\leq \exp\left(2x\eta^2 r_{t_0}^2 \cdot \frac{1}{1 - e^{-\Theta(2\mu\eta-1)}}\right) \cdot s_{t_0}$$

$$\leq \exp\left(\frac{2x\eta^2 r_{t_0}^2}{\Theta(2\mu\eta - 1)}\right) \cdot s_{t_0}$$

Since $|r_{t_0}| \leq \mu$ and $s_{t_0} = \eta c_x \cdot b_{t_0} \leq \mu^{-\Theta(1)} \cdot \alpha^{\Theta(1)}$ (due to Lemma 7), we conclude that

$$\lim_{t \to \infty} s_t \leq \exp\left(\frac{2x\eta^2\mu^2}{\Theta(2\mu\eta - 1)}\right) \cdot s_{t_0} = \exp\left(\frac{2x}{\Theta(2\mu\eta - 1)}\right) \cdot s_{t_0} = O(\alpha^{\Theta(1)}),$$

where we hide the dependence on $\eta, \mu, x$ because we want to focus on the asymptotic property on $\alpha$ only. $\qquad\square$

Finally, we put every piece together and prove the result for $\eta\mu < 1$.

*Proof of Theorem 1.* We first show part (1) is true. From Lemma 4, Lemma 5, we can decompose $\boldsymbol{\beta}_{\boldsymbol{w}_t}$ as

$$\boldsymbol{\beta}_{\boldsymbol{w}_t} = \boldsymbol{w}_{t,+}^2 - \boldsymbol{w}_{t,-}^2 = (p_t^0 - q_t^0) \cdot \boldsymbol{\beta}_0 + (p_t^1 - q_t^1) \cdot \boldsymbol{\beta}_1 + \boldsymbol{\beta}_*$$

where the vectors are $\boldsymbol{\beta}_0 = (1, x)$, $\boldsymbol{\beta}_1 = (x, -1)$ and the coefficients can be computed as

$$p_t^0 - q_t^0 = a_t - a_t' - \frac{\mu}{1 + x^2} + x \cdot (b_t - b_t'),$$

$$p_t^1 - q_t^1 = -x \cdot \left(a_t - a_t' - \frac{\mu}{1 + x^2}\right) + b_t - b_t'.$$

We now proceed to discuss the limits of the coefficients for $\boldsymbol{\beta}_0$ and $\boldsymbol{\beta}_1$.

We notice it suffices to consider $x > 0$: by observing Lemma 5, the iteration with $x = -a$, $a > 0$ can be regarded as simply exchanging $b_t/b_t'$ of an iteration with $x = a$. Starting from initialization

$$a_0 = a_0' = b_0 = b_0' = \frac{\alpha^2}{2(1 + x^2)}$$

(which corresponds to $\boldsymbol{w}_{t,\pm} = \alpha\mathbf{1}$), Lemma 7 suggests that $a_t'$ and $b_t'$ will decrease and converge to $\frac{\alpha^2}{2(1+x^2)}$ and $0$ very quickly. In the meanwhile, $a_t$, although remains negative, increases to the level of $-\Theta(\mu)$. Therefore, we can simplify the analysis by discard early iterations and regarding $a_t', b_t'$ as constantly $0$, which leaves us a discrete dynamical system of two variables $(a_t, b_t)$ and it holds that $r_t = (1 + x^2) \cdot (a_t + x \cdot b_t) - \mu$ for any $t \geq t_0$, where $t_0$ is by our assumption the start of change of sign.

We notice the following inner products

$$\langle \boldsymbol{\beta}_0, \boldsymbol{x} \rangle = 1 + x^2, \qquad \langle \boldsymbol{\beta}_1, \boldsymbol{x} \rangle = 1, \qquad \text{and} \qquad \langle \boldsymbol{\beta}^*, \boldsymbol{x} \rangle = \mu. \tag{13}$$

This allows us to express $r_t$ as

$$r_t = \langle \boldsymbol{\beta}_{\boldsymbol{w}_t}, \boldsymbol{x} \rangle - \mu = (p_t^0 - q_t^0) \cdot \langle \boldsymbol{\beta}_0, \boldsymbol{x} \rangle + (p_t^1 - q_1^0) \cdot \langle \boldsymbol{\beta}_0, \boldsymbol{x} \rangle + \langle \boldsymbol{\beta}^*, \boldsymbol{x} \rangle - \mu$$

$$= (1 + x^2) \cdot (p_t^0 - q_t^0)$$

We now invoke Lemma 10 and obtain

$$\lim_{t\to\infty} p_t^0 - q_t^0 \propto \lim_{t\to\infty} r_t = 0,$$

as well as

$$
\begin{aligned}
\lim_{t\to\infty} p_t^1 - q_t^1 &= \lim_{t\to\infty} \left( -x \cdot \left( a_t - a_t' - \frac{\mu}{1+x^2} \right) + b_t - b_t' \right) \\
&= -x \cdot \lim_{t\to\infty} \left( (a_t - a_t') + b_t - b_t' \right) \\
&= -x \cdot \lim_{t\to\infty} r_t + x^2 \cdot \lim_{t\to\infty} (b_t - b_t') = x^2 \cdot \lim_{t\to\infty} (b_t - b_t') \leq O(\alpha^C).
\end{aligned}
$$

where $C > 0$ is some constant. The above results suggest that $\boldsymbol{\beta}_{\boldsymbol{w}_t}$ converges to the following limit

$$\boldsymbol{\beta}_\infty := \lim_{t\to\infty} \left( p_t^0 - q_t^0 \right) \cdot \boldsymbol{\beta}_0 + \left( p_t^1 - q_t^1 \right) \cdot \boldsymbol{\beta}_1 + \boldsymbol{\beta}^* = \boldsymbol{\beta}^* + \boldsymbol{\beta}_1 \cdot x^2 \cdot \lim_{t\to\infty} b_t - b_t'.$$

It is easy to verify $\boldsymbol{\beta}_\infty$ is a linear interpolator $\langle \boldsymbol{\beta}_\infty, \boldsymbol{x} \rangle = y$ due to (13). Therefore, the convergence to $\boldsymbol{\beta}_\infty$ can be characterized in a non-asymptotic manner using Lemma 10 again: for any $t \geq t_0$,

$$|\langle \boldsymbol{\beta}_{\boldsymbol{w}_t} - \boldsymbol{\beta}_\infty, \boldsymbol{x} \rangle| = |\langle \boldsymbol{\beta}_{\boldsymbol{w}_t} - \boldsymbol{\beta}^*, \boldsymbol{x} \rangle| = |r_t| \leq C \cdot \exp\left( -\Theta(2\mu\eta - 1) \cdot (t - t_0) \right) |r_{t_0}|.$$

Moreover, it holds that

$$\|\boldsymbol{\beta}_\infty - \boldsymbol{\beta}^*\| = \lim_{t\to\infty} (p_t^1 - q_t^1) \cdot \|\boldsymbol{\beta}_1\| = O(\alpha^C).$$

$\square$

## A.4 Proof of Theorem 2: Regime $\eta\mu \in (1, \frac{3\sqrt{2}-2}{2})$

This subsection contains the EoS convergence proof under the much harder case with $\eta\mu > 1$. We continue to use the $(r_t, s_t)$ update in Appendix A.3.

**Proof outline.** Compared with the regime $\eta\mu < 1$, it poses more challenging tasks when we attempt to provide a convergence proof under $\eta\mu > 1$. This is because, by the end of the initial phase, the iterate of $r_t$ does not start to contract. On the contrary, the envelope of the oscillating $r_t$ can even increase, which suggests $|r_{t+2}| > |r_t|$ for some certain $t$. Therefore, in contrast to the setting $\mu\eta < 1$, we can not directly apply Lemma 9.

We observe Lemma 9 again and realize that the convergence is decided by the following criterion

$$\alpha_t \geq 0, \quad \text{or equivalently,} \quad s_t \geq \mu\eta - 1 \quad \Longrightarrow \quad \text{contraction of } |r_t| \text{ happens.}$$

Recall that $\alpha_t = 2 - 2\eta\mu + 2s_t$. From the discussion on the initial phase, it is shown that $s_t$ (or equivalently $b_t$) starts with a small value at $t_0$, which implies in early phases $\alpha_t < 0$ and the contraction criterion is not satisfied. This is why convergence does not happen in the second phase in the case of $\mu\eta > 1$. Therefore, the central task is to find under which condition $b_t$ and $s_t$ increase sufficiently during the second phase such that the phase transition from $\alpha_t < 0$ to $\alpha_t > 0$ will eventually occur.

Through a fixed-point analysis of the discrete dynamical system, we are able to show that phase transition will happen under $x \in (0, \frac{1}{\mu\eta})$, and subsequently we could employ Lemma 9 to establish the convergence.

**Auxiliary sequence.** Before we prove the phase transition will occur when $x \in (0, \frac{1}{\mu\eta})$, it requires characterizing some properties of the $(r_t, s_t)$ iterate using the tool of discrete dynamical system and an auxiliary sequence defined as

$$\rho_{t+1} = g(\rho_t, 0)$$

with initialization $\rho_{t_0} = r_{t_0}$. The new sequence can be regarded as setting $s_t \equiv 0$ for any $t$. We start by presenting some basic properties of $\rho_t$ sequence and mapping $g(\cdot, 0)$.

**Lemma 11.** *The following statements are true for the iteration of $\rho_t$:*

*(1). The local minimum and maximum of $g(\rho, 0)$ are $\rho = \frac{1}{\eta}$ and $\rho = -\frac{2\eta\mu-1}{3\eta}$, respectively. As a result, the mapping $g(r, 0)$ is monotonically decreasing and sign-changing ($\rho \cdot g(\rho, 0) \le 0$) in the range of $\rho \in [-\frac{2\eta\mu-1}{3\eta}, \frac{1}{\eta}]$;*

*(2). The mapping $g(\rho, 0)$ has 2-periodic points at $\rho_\pm = \frac{1-\mu\eta \pm \sqrt{\mu\eta^2 + 2\mu\eta - 3}}{2\eta}$.*

*Proof.* To prove (1), we take derivative of $g(\rho, 0)$ with respect to the first variable:

$$g'(\rho, 0) = 3\eta^2\rho^2 + 2(\mu\eta^2 - 2\eta)\rho - 2\mu\eta + 1.$$

Because $g(\rho, 0)$ is a degree-3 polynomial in $\rho$, solving equation $g'(\rho, 0) = 0$ suggests that $\frac{1}{\eta}$ and $-\frac{2\eta\mu-1}{3\eta}$ are the only two stationary points of $g(\cdot, 0)$. Moreover, we compute the value and derivative at $\rho = 0$ as

$$g(0, 0) = 0, \qquad g'_\rho(0, 0) = 1 - 2\mu\eta < 0,$$

as a result, $g(\rho, 0) \cdot \rho < 0$ holds and $g(\rho, 0)$ is monotonically decreasing when $\rho \in [-\frac{2\eta\mu-1}{3\eta}, \frac{1}{\eta}]$.

To prove (2), we solve fixed-point equation $r = g(g(r, 0), 0)$ to obtain a pair of non-trivial solutions:

$$\rho_\pm = \frac{1 - \mu\eta \pm \sqrt{\mu^2\eta^2 + 2\mu\eta - 3}}{2\eta}.$$

$\square$

When $\eta\mu \in (1, \frac{3\sqrt{2}-2}{2})$, the sequence $\rho_t$ can be regarded as a "reference" of $r_t$ because its envelope contains $r_t$'s envelope. This means $r_t, \rho_t$ have the same sign and $|r_t| \le |\rho_t|$ holds for any $t$ if they share a proper initialization. This is stated in the next lemma.

**Lemma 12.** *Consider the sequence $(r_t, s_t)$ and $\rho_t$ with same initialization $r_{t_0} = \rho_{t_0} \in (-\frac{\mu}{2}, 0)$ and $s_{t_0} \ge 0$. If $\eta\mu \in \left(1, \frac{3\sqrt{2}-2}{2}\right)$ and $r_t r_{t+1} < 0$ holds for any $t \ge t_0$, then the following statements are true for any $t \ge t_0$: (1) $r_t$ and $\rho_t$ have the same sign, (2) $|r_t| \le |\rho_t|$ and (3) $r_t, \rho_t \in [-\rho_-, \rho_+]$.*

*Proof.* Let $\eta\mu \in \left(1, \frac{3\sqrt{2}-2}{2}\right)$. Our first goal is to show that with initialization $\rho_{t_0} \in [\rho_-, \rho_+]$, where $\rho_\pm$ are the 2-periodic points, then for any $t$, $\rho_t \rho_{t-1} < 0$ and $r_t \in [\rho_-, \rho_+]$ hold. We prove this by induction. Now suppose this holds for $t$. It is easy to check when $\eta\mu \in \left(1, \frac{3\sqrt{2}-2}{2}\right)$, the following inequalities are true:

$$-\frac{2\eta\mu-1}{3\eta} \le \frac{1-\mu\eta - \sqrt{\mu^2\eta^2 + 2\mu\eta - 3}}{2\eta} < 0 < \frac{1-\mu\eta - \sqrt{\mu^2\eta^2 + 2\mu\eta - 3}}{2\eta} < \frac{1}{\eta}.$$

Therefore, the mapping $g(\rho, 0)$ is monotonically decreasing and sign-changing from (1) of Lemma 11. As a result, when $\rho_t < 0$, we have

$$0 \le \rho_{t+1} = g(\rho_t, 0) < g(\rho_-, 0) = \rho_+.$$

Similarly, when $\rho_t > 0$, we have

$$0 \ge \rho_{t+1} = g(\rho_t, 0) \ge g(\rho_+, 0) = \rho_-.$$

Therefore (1) and $\rho_t$ part in (3) are proved. It remains to prove for (2) which immediately implies $r_t \in [\rho_-, \rho_+]$ in (3). Still, we prove this by induction on $t$. Suppose that $|r_t| \le |\rho_t|$ is true for $t$. When $r_t < 0$,

$$r_{t+1} = g(r_t, s_t) = g(r_t, 0) + r_t s_t \cdot \left(2 - (1 + x^2)\eta r_t\right) \le g(r_t, 0) \le g(\rho_t, 0),$$

where the last inequality comes from the monotonicity of $g(\cdot, 0)$ and condition $r_t \in [\rho_-, \rho_+]$. When $r_t > 0$, it holds that

$$r_{t+1} = g(r_t, 0) + 2r_t s_t - (1 + x^2)\eta r_t^2 s_t$$
$$\ge g(r_t, 0) + 2r_t s_t - (1 + x^2)r_t s_t \ge g(r_t, 0) \ge g(\rho_t, 0),$$

where the first inequality comes from $r_t \le \rho_- \le \frac{1}{\eta}$ and the second is from $x \le 1$. $\square$

Finally, we give the condition on when the bivariate mapping $(g, h)$ admits periodic points.

**Lemma 13** (2-periodic points)**.** *If $x \in (0, \frac{1}{\eta\mu})$, then the mapping $(g, h) : \mathbb{R}^2 \to \mathbb{R}^2$ does not admit 2-periodic points, and hence also $2^n$-periodic points for $n \geq 1$, in the domain $(\mathbb{R} \setminus \{0\}) \times (0, \mu\eta - 1)$. On the contrary, if $x \in (\frac{1}{\eta\mu}, 1)$, $(g, h)$ admits a pair of non-trivial 2-periodic points $(r_\pm, s_\pm)$ in the same domain as*

$$r_\pm = \frac{1 - \mu\eta x \mp \sqrt{\mu^2\eta^2 x^2 + 2\mu\eta x - 3}}{2\eta x},$$

$$s_\pm = \frac{(1 - x)(\mu\eta x + 1 \pm \sqrt{(\mu\eta x + 1)^2 - 4})}{2x}.$$

*Proof.* Before proceeding to the discussion 2-periodic points, we compute the fixed point, a.k.a. 1-periodic points of $(g, h)$. This is because every 1-periodic point is also a trivial 2-periodic point which we need to eliminate. To obtain the fixed points, it requires to solve the following equation system

$$\begin{cases} g(r, s) = r, \\ h(r, s) = s. \end{cases}$$

Clearly $(r_{1,1}, s_{1,1})$ with $r_{1,1} = 0$ and any $s_{1,1} \in \mathbb{R}$ is a trivial solution. We employ Mathematica Symbolic Calculation to obtain another root $(r_{1,2}, s_{1,2}) = \left( \frac{2}{\eta x}, \frac{(1-x)(2+x\mu\eta)}{x} \right)$.

Now we compute the non-trivial 2-periodic points by solving the following equation system

$$\begin{cases} g(g(r, s), h(r, s)) = r, \\ h(g(r, s), h(r, s)) = s. \end{cases}$$

The computation result is also obtained by Mathematica Symbolic Calculation. We compare with the above fixed point, which indicates the first three real roots

$$(r_{2,1}, b_{2,1}) = (0, \mu\eta), \quad (r_{2,2}, b_{2,2}) = (0, \mu\eta - 1), \quad \text{and} \quad (r_{2,3}, b_{2,3}) = \left( \frac{2}{\eta x}, \frac{(1-x)(2+x\mu\eta)}{x} \right)$$

are also the 1-periodic points and hence trivial. There are four pairs of possibly real roots $(r_{2,i,\pm}, s_{2,i,\pm})$ for $i = \{4, 5, 6, 7\}$ as following:

$$r_{2,4,\pm} = \frac{x + 1 \mp \sqrt{x^2 + 1}}{\eta x},$$

$$s_{2,4,\pm} = (\mu\eta x + x^2 + x + 2) \left( \frac{1}{x} \pm \frac{1}{\sqrt{1 + x^2}} \right),$$

$$r_{2,5,\pm} = \frac{x \mp \sqrt{x^2 - 2x + 2}}{\eta x - \eta},$$

$$w_{2,5,\pm} = \frac{(\mu\eta(x - 1) + x^2 - x + 2)(x^2 - x + 1 \pm x\eta\sqrt{x^2 - 2x + 2})}{(x - 1)^3},$$

$$r_{2,6,\pm} = \frac{x - \mu\eta x^2 \mp x\sqrt{\mu^2\eta^2 x^2 + 2\mu\eta x - 3}}{2\eta x^2},$$

$$s_{2,6,\pm} = \frac{(1 - x)(\mu\eta x + 1 \pm \sqrt{\mu^2\eta^2 x^2 + 2\mu\eta x - 3})}{2x},$$

and

$$r_{2,7,\pm} = \frac{1 \pm \sqrt{2x^2 - 2x + 1}}{\eta(1 - x)x},$$

$$s_{2,7,\pm} = \frac{\left(1 + 2x - 2x^2 \mp x\sqrt{1 + 2x - 2x^2}\right)(x(1 - x)(\mu\eta - 1) + 2)}{(1 - x)^3 x}.$$

It is easy to verify that for any $i \in \{4, 5, 6, 7\}$, it holds that

$$g(r_{2,i,\pm}, s_{2,i,\pm}) = r_{2,i,\mp}, \qquad h(r_{2,i,\pm}, s_{2,i,\pm}) = s_{2,i,\mp},$$

which suggests these pairs are indeed 2-periodic points if they are real number given condition $s \in (0, \mu\eta - 1)$.

Now we discuss if the points are legitimate by considering the constraint $s \in (0, \mu\eta - 1)$. We first notice that

$$
\begin{aligned}
s_{2,5,+} + s_{2,5,-} &= -\frac{(x^2 - x + 1)\left(\mu\eta(x-1) + x^2 - x + 2\right)}{(1-x)^3} \\
&= -\frac{(x^2 - x + 1)(2 - (\mu\eta - x)(1-x))}{(1-x)^3} \\
&\leq -\frac{\left((x - \frac{1}{2})^2 + \frac{3}{4}\right)\left(2 - (2-x)(1-x)\right)}{(1-x)^3} \\
&= -\frac{\left((x - \frac{1}{2})^2 + \frac{3}{4}\right)\left(3x - x^2\right)}{(1-x)^3} < 0
\end{aligned}
$$

due to the fact that $x \in (0,1)$ and $\eta\mu \in [0,2]$. This indicates at least one of $s_{2,5,\pm}$ must be negative and this pair should also be discarded. Also, we discard the pair $(r_{2,4,\pm}, s_{2,4,\pm})$ and $(r_{2,7,\pm}, s_{2,7,\pm})$ because

$$
s_{2,4,+} > \frac{2}{x} + x \geq 2\sqrt{2} > \mu\eta - 1,
$$

and

$$
s_{2,7,-} > \frac{x(1-x)(\mu\eta + 1)}{x(1-x)^3} = \frac{\eta\mu + 1}{(1-x)^2} > \eta\mu - 1.
$$

It remains to investigate the last pair of $(r_{2,6,\pm}, s_{2,6,\pm})$. We notice the following identity

$$
\mu^2\eta^2 x^2 + 2\mu\eta x - 3 = (\mu\eta x + 1)^2 - 4.
$$

As a result, if $x \in (0, 1/(\mu\eta))$, it holds that $(\mu\eta x + 1)^2 - 4 < 0$ and hence the roots are complex. On the contrary, if $x \in (1/(\mu\eta), 1)$, it holds that $(\mu\eta x + 1)^2 - 4 > 0$ and hence it admits real roots. Therefore we conclude that when $x \in (0, 1/(\mu\eta))$, there is no 2-periodic points for the mapping $(g, h)$; in the meanwhile, a pair of 2-periodic points exists when $x \in (1/(\mu\eta), 1)$.

We can verify there are four additional roots that have $\sqrt{-7 + 2\mu\eta x + \mu^2\eta^2 x^2}$ in the fractional and hence are always complex because $-7 + 2\mu\eta x + \mu^2\eta^2 x^2 < 0$ for any $\mu\eta \in [0,2]$ and $x \in [0,1]$. Therefore they do not constitute 2-periodic points in the real domain. $\qquad\square$

**Phase transition and thresholding-crossing.** With the above lemmas featuring the $(r_t, s_t)$ iterations, we now prepare to prove the phase transition from $\alpha_t < 0$ to $\alpha_t > 0$ will eventually happen, turning the envelope of $r_t$ to convergent. It should be noticed that $\alpha \lessgtr 0$ is equivalent to $s_t \lessgtr \mu\eta - 1$, which we use interchangeably.

**Lemma 14.** *Suppose that $\eta\mu \geq (1, \frac{3\sqrt{2}-2}{2}]$ and $x \in (0, \frac{1}{\mu\eta})$. Consider $r_{t_0} \in (-\frac{\mu}{2}, 0)$ and $0 < s_{t_0} \ll \mu\eta - 1$. Then it holds that $\lim_{t\to\infty} s_t > \mu\eta - 1$, or equivalently $\lim_{t\to\infty} \alpha_t > 0$.*

*Proof.* We consider the behavior of the discrete system of $(r_t, s_t)$ described in (11). Asymptotically, it either diverges, becomes chaotic, or converges to a fixed point or a periodic cycle. Let us suppose that $\limsup_{t\to\infty} \alpha_t < 0$, which is equivalent to saying $\limsup_{t\to\infty} s_t < \mu\eta - 1$. We will prove later that under parameter choice $x \in (0, \frac{1}{\eta\mu})$, the iterates $(r_t, s_t)$ do not diverge, become chaotic, nor converge to a periodic orbit. As a result, either $\alpha_t$ will finally come across **zero**, or $\limsup_{t\to\infty} \alpha_t > 0$ still holds but $\lim_{t\to\infty} r_t = 0$. Since the total measure of the latter event is negligible, we only take the thresholding-crossing case into account.

Now we analyze the behavior of $(r_t, s_t)$ when the crossing of $\alpha_t$ does not occur. We consider the auxiliary iteration $\rho_{t+1} = g(\rho_t, 0)$ with the same initialization at $r_{t_0}$ and prove the following statements are true for any $t \geq t_0$:

$$
(1) \quad r_t r_{t+1} < 0, \quad \text{and} \quad (2) \quad |r_t| \in |\rho_t|.
$$

This is proved by induction over $t \geq t_0$. For the base case $t = t_0$, it can be easily verified from

$$r_0 \geq \frac{\mu}{2} > \frac{1 - \mu\eta - \sqrt{\eta^2\mu^2 + 2\eta\mu - 3}}{2\eta} \geq -\frac{\mu\eta + 1}{2\eta}.$$

Therefore, the assumption $r_{t_0} \in (-\mu/2, 0)$ in Lemma 12 is met and allows us to use the lemma once change of sign is proved. Now suppose they hold for any $t - 1$. It is immediately from the update of $r_t$

$$-\frac{r_{t+1}}{r_t} = 1 - \alpha_t + \beta_t r_t.$$

To establish (1), it suffices to show $1 - \alpha_t + \beta_t r_t \geq 1 + \beta_r r_t > 0$ due to condition $\alpha_t < 0$. We discuss two separate cases: $r_t \leq 0$ and $r_t > 0$. In the first case of $r_t \leq 0$, we write down

$$1 + \beta_t r_t = 1 + 2\eta r_t - \eta^2 r_t \cdot \left(\mu + r_t - c_x' b_t\right)$$
$$= 1 + 2\eta r_t - \eta^2 r_t \cdot (1 + x^2) \cdot \left(a_t + x^3 \cdot b_t\right)$$
$$= 1 + 2\eta r_t.$$

where in the first line we use the definition of $\beta_t$, in the second line we use the identity in Lemma 5, and in the last line we use facts $a_t, b_t \geq 0$. By our assumption, $r_t r_{t-1} < 0$ is true and hence we can invoke Lemma 12 to conclude that $r_t \geq \rho_-$. This leads to

$$-\frac{r_{t+1}}{r_t} \geq 1 + 2\eta r_t \geq 1 + 2\eta\rho_- \geq 1 - 2\eta \cdot \frac{2\eta\mu - 1}{3\eta} = \frac{5}{3} - \frac{4}{3}\mu\eta > 0.$$

because $\eta\mu \leq \frac{3\sqrt{2} - 2}{2} < 5/4$. In the second case of $r_t \geq 0$, we have

$$1 + \beta_t r_t = 1 + 2\eta r_t - \eta^2\mu r_t - \eta^2 r_t^2 + \eta^2 c_x' b_t r_t$$
$$\geq 1 + \eta r_t \cdot \left(2 - \eta\mu - \eta r_t\right).$$

Similar to the above discussion, since $r_t r_{t-1} < 0$, we are able to obtain $r_t \leq \rho_+$ using Lemma 12. As a result, we have

$$-\frac{r_{t+1}}{r_t} \geq 1 + \eta r_t\left(2 - \eta\mu - \eta/\eta\right) = 1 - \eta r_t\left(\eta\mu - 1\right) \geq 2 - \eta\mu > 0,$$

where the second line comes from $r_t > 0$ and $b_t > 0$. Summarizing both cases, we reach the conclusion $r_t r_{t+1} < 0$. Since the condition is met, we can invoke Lemma 12 to conclude that $r_{t+1} \in [\rho_-, \rho^+]$ is also true. With the above statements to be true, the $r_t$ iteration is bounded and not diverging. Also, because $x \in (0, \frac{1}{\eta\mu})$, Lemma 13 indicates that $(r_t, s_t)$ does not converge to any $2^n$-periodic orbit or become chaotic by bifurcation theory. This proves the existence of $\mathsf{t}$ and the final result. $\qquad\square$

The above lemma suggests that there exists a certain $\mathsf{t}$ such that $\alpha_t < 0$ should hold for any $t \geq \mathsf{t}$, which marks the beginning of the *convergence* phase. We next characterize the properties and, in particular, give an upper bound estimate for $\alpha_t$ when the transition happens through the following lemma for the initial gap. These results are important when we establish the convergence rate in this phase.

**Lemma 15.** *Suppose that $\eta\mu \geq 1$ and $x \in (0, \frac{1}{\mu\eta})$. Then there exists a $\mathsf{t}$ such that $0 \leq \alpha_t$ is true for any $t \geq \mathsf{t}$. Moreover, it holds that $r_\mathsf{t} > 0$ and $\alpha_\mathsf{t} \leq \Theta(\mu\eta - 1)$.*

*Proof.* By Lemma 14, $\lim_{t \to \infty} \alpha_t > 0$ should hold under $x \in (0, \frac{1}{\eta\mu})$, which suggests there exists some $t \in \mathbb{R}$ such that $\alpha_{t'} > 0$ holds for any $t' \geq t$. Let $\mathsf{t}$ be the least of such $t$'s. We first prove that $r_\mathsf{t} > 0$. Let us suppose not, hence we have $r_\mathsf{t} < 0$ and $r_{\mathsf{t}-1} > 0$. This indicates that

$$s_{\mathsf{t}-1} = (1 - x\eta r_{\mathsf{t}-1})^{-2} \cdot s_\mathsf{t} > s_\mathsf{t}$$

and therefore $\alpha_{\mathsf{t}-1} > \alpha_\mathsf{t} > 0$. Hence $\mathsf{t}$ is not the least $t$ such that $\alpha_{t'} > 0$ holds for any $t' \geq t$.

We proceed and upper bound $\alpha_\mathsf{t}$. Suppose that $r_{\mathsf{t}-1} \geq -\mu$ holds and we will postpone its proof. Under this condition, we upper bound $s_\mathsf{t}$ as:

$$s_\mathsf{t} = (1 - x\eta r_{\mathsf{t}-1})^2 \cdot s_{\mathsf{t}-1} \leq (\mu\eta - 1) \cdot (1 - x\eta r_{\mathsf{t}-1})^2 \leq (\mu\eta - 1) \cdot (1 + x\eta\mu) \leq 4(\mu\eta - 1),$$

where the last inequality is due to $x \in (0, \frac{1}{\mu\eta})$. Therefore $\alpha_\mathsf{t} = 2s_\mathsf{t} - 2(\mu\eta - 1) \leq 2(\mu\eta - 1)$. Finally, we argue that $r_{\mathsf{t}-1} \in (-\mu, 0)$. Because $a_t > 0$, $b_t > 0$ holds for any $t$, from the expression of $r_t$ we deduce that $r_t = (1 + x^2) \cdot (a_t + x \cdot b_t) - \mu > -\mu$. $\qquad\square$

**Lemma 16.** *Suppose that $\alpha_t > 0$, $s_t > 0$, $r_t < 0$ and $x \in (0, \frac{1}{\mu\eta})$. Then it holds that $(1 - x\eta r_t)(1 - x\eta r_{t+1}) \geq 1$.*

*Proof.* We expand the term using Eq. (10)

$$
\begin{aligned}
(1 - x\eta r_t)(1 - x\eta r_{t+1}) &= (1 - x\eta r_t) \cdot \left(1 - x\eta(1 - \alpha_t + \beta_t r_t) \cdot r_t\right) \\
&= (1 - x\eta r_t) \cdot (1 - x\eta r_t) + x\eta\alpha_t r_t \cdot (1 - x\eta r_t) + x\eta\beta_t r_t^2 \cdot (1 - x\eta r_t) \\
&\geq 1 - x^2\eta^2 r_t^2 + x\eta\beta_t r_t^2 \\
&\geq 1
\end{aligned}
$$

where the first inequality is from $r_t < 0$, $\alpha_t > 0$. For the second inequality, we calculate

$$
\begin{aligned}
\beta_t &= 2\eta - \eta^2\left(\mu + r_t - (1-x)s_t\right) \\
&= 2\eta - \eta^2\mu - \eta^2 r_t + (1-x)\eta^2 s_t) \\
&\geq 2\eta - \eta^2\mu = \eta(2 - \eta\mu) \geq x\eta
\end{aligned}
$$

due to $r_t < 0$, $s_t > 0$ and $x\eta\mu \leq 1$. □

**Convergence phase.** The above lemmas indicate if $x \in (0, \frac{1}{\eta\mu})$ is true, the iteration will finally enter the third phase, where $\alpha_t > 0$ guarantees the convergence.

We state the result in the next Lemma, where we establish the convergence of $r_t$ using Lemma 9, very similar to Lemma 10 in Appendix A.3.

**Lemma 17.** *Suppose that $x \in (0, \frac{1}{\eta\mu})$. Then there exists a universal constant $C > 0$ such that for any $\mu\eta \in (1, \min\{\frac{3\sqrt{2}-2}{2}, 1 + C^{-1}/4\})$ and any $t \geq \mathfrak{t}$, (a) $r_t r_{t+1} < 0$ and (b) the iteration $(r_t, s_t)$ converges to $(0, s_\infty)$ in a linear rate as*

$$|r_t| \leq \exp\left(-\Theta(\mu\eta - 1) \cdot (t - \mathfrak{t})\right) \cdot |r_{\mathfrak{t}}|.$$

*Moreover, it holds that $\lim_{t\to\infty} b_t \leq C(\mu\eta - 1)$.*

*Proof.* Lemma 15 states that $r_{\mathfrak{t}} > 0$, which suggests $r_t < 0$ holds if $t - \mathfrak{t}$ is an odd number. Therefore, we prove the lemma by considering any $t$ with $t - \mathfrak{t}$ to be odd: suppose for any such $t$, the following statements are true:

(1). $r_{t+1} > 0$, $r_{t+2} < 0$;

(2). $|r_{t+1}| \leq O(|r_t|)$, $|r_{t+2}| \leq (1 - \Theta(\mu\eta - 1))^2 \cdot |r_t|$;

(3). $s_t \leq s_{t+2} \leq C(\mu\eta - 1)$.

Then we are able to the change of sign in (a) is true for any consecutive iteration. In the meanwhile, we can repeatedly use (2) to establish the non-asymptotic linear convergence of $|r_t|$ as well as the uniform upper bound of $s_t$ for any $t \geq \mathfrak{t}$. We will prove the correctness using induction. Suppose the above statements are true for $t' \leq t$ with $t - t'$ to be even, given that $t - \mathfrak{t}$ is also even. Then $r_t < 0$ holds. We first show (1) is true. Similar to the proof of Lemma 14, it suffices to show $1 - \alpha_t + \beta_t r_t > 0$. We can expand the term by plugging the definition of $\beta_t$

$$1 - \alpha_t + \beta_t r_t = 1 + \alpha_t + 2\eta r_t - \eta^2 r_t \cdot \left(\mu + r_t - c'_x b_t\right) \geq 1 + \alpha_t + 2\eta r_t.$$

where the inequality is due to $r_t < 0$. Moreover, we insert the definition of $\alpha_t$

$$1 + \alpha_t + 2\eta r_t = 1 - 2 + 2\eta\mu - 2s_t + 2\eta r_t.$$

We consider the auxiliary sequence $\rho_{t+1} = g(\rho_t, 0)$ with the same initialization at $r_{t_0}$. Using the argument in the proof of Lemma 14 and our condition $r_{t'} r_{t'+1} < 0$ for any $t' \leq t - 1$, we are able to invoke Lemma 12 to obtain $r_t \geq \rho_-$. This leads to a lower bound

$$
\begin{aligned}
1 + \alpha_t + 2\eta r_t &\geq 1 - 2 + 2\eta\mu - 2s_t + 2\eta\rho_- \\
&\geq 1 + 2(\eta\mu - 1) - 2C(\mu\eta - 1) - 3(\mu\eta - 1) \\
&= 1 - (2C + 1)(\eta\mu - 1) > 0.
\end{aligned}
$$

due to (3), $\rho_- = -\frac{\mu\eta - 1 + \sqrt{(\mu\eta-1)(\mu\eta+3)}}{2\eta} \geq -\frac{3(\mu\eta-1)}{2\eta}$, and $\eta\mu \in (1, \min\{\frac{3\sqrt{2}-2}{2}, 1 + 1/(4C)\})$. This immediately implies $r_t r_{t+1} < 0$ and $r_{t+1} > 0$. For the sign of $r_{t+2}$, we first notice

$$\beta_{t+1} = 2\eta - \eta^2 \cdot (r_t + \mu - c'_x b_t) = 2\eta - \eta^2 \cdot (1 + x^2) \cdot (a_t + x^3 \cdot b_t) < 2\eta$$

due to the non-negativity of $a_t$ and $b_t$. Besides, because $\rho_- \leq r_t < 0$, we have

$$(1 - x \cdot \eta r_t)^2 \leq (1 - x \cdot \eta\rho_-)^2 \leq \left(1 + \frac{3(\mu\eta - 1)}{2}\right)^2 < 2$$

where the last inequality is due to $\mu\eta \leq \frac{3\sqrt{2}-2}{2}$. As a result, we lower bound as

$$1 - \alpha_{t+1} + \beta_{t+1} r_{t+1} \geq 1 - \alpha_{t+1} = 1 - 2 + 2\mu\eta - 2s_{t+1}$$
$$= 1 - 2 + 2\mu\eta - (1 - x \cdot r_t)^2 \cdot 2s_t$$
$$\geq 1 - (4C - 2)(\mu\eta - 1) > 0.$$

This immediately yields $r_{t+2} r_{t+1} > 0$ and hence $r_{t+2} < 0$.

For (2), since $r_t r_{t+1} < 0$ is true, we invoke Lemma 9 to obtain the lower bound for $r_{t+2}$, which is also negative by our assumption:

$$r_{t+2} \geq (1 - \alpha_t)(1 - \alpha_{t+1}) \cdot r_t.$$

Because $r_t < 0$, $\alpha_t > 0$ and Lemma 16, it holds that

$$\alpha_{t+1} = 2 - 2\eta\mu + 2s_{t+1} \geq 2 - 2\eta\mu + 2s_{\mathsf{t}+1} \geq 2 - 2\eta\mu + 2s_{\mathsf{t}+1},$$

which suggests $\alpha_{t+1} \geq \alpha_t \geq \alpha_{\mathsf{t}+1} = \Theta(\mu\eta - 1)$. As a result

$$|r_{t+2}| \leq (1 - \Theta(\mu\eta - 1))^2 \cdot |r_t|.$$

From the above discussion, we know $1 - \alpha_t + \beta_t r_t > 0$ and $\beta_t > 0$. Then we compute

$$\frac{|r_{t+1}|}{|r_t|} = 1 - \alpha_t + \beta_t r_t \leq 1 - 2 + 2\mu\eta = 2\mu\eta - 1 \leq 2$$

due to $\mu\eta \leq 2$ and $r_t < 0$. This implies $|r_{t+1}| \leq O(|r_t|)$. It remains to check for (3). We first prove the first half, i.e,

$$s_{t+2} = (1 - x\eta r_t)^2 (1 - x\eta r_{t+1})^2 \cdot s_t \geq s_t$$

where the inequality is due to $r_t < 0$, $\alpha_t \geq 0$ and Lemma 16. By repeating the above steps, we conclude that $s_t \geq s_{\mathsf{t}+1}$ for any $t \geq \mathsf{t}$ with $r_t < 0$. For the upper bound of $s_{t+2}$, we skip some calculations identical to the proof of Lemma 10 and obtain

$$s_t = \exp\left(2x\eta^2 \cdot \sum_{\substack{i=\mathsf{t} \\ i-\mathsf{t}\text{ even}}}^{t-2} r_t^2\right) \cdot s_{\mathsf{t}} \leq \exp\left(2x\eta^2 r_{\mathsf{t}}^2 \cdot \sum_{\substack{i=\mathsf{t} \\ i-\mathsf{t}\text{ even}}}^{t-2} e^{-\Theta(\mu\eta-1)\cdot(t-\mathsf{t})}\right) \cdot s_{\mathsf{t}}$$

$$\leq \exp\left(2x\eta^2 r_{\mathsf{t}}^2 \cdot \sum_{\substack{i=\mathsf{t} \\ i-\mathsf{t}\text{ even}}}^{t-2} \frac{1}{1 - e^{-\Theta(\mu\eta-1)\cdot(t-\mathsf{t})}}\right) \cdot s_{\mathsf{t}}$$

$$\leq \exp\left(\frac{2x\eta^2 r_{\mathsf{t}}^2}{\Theta(\mu\eta - 1)}\right) \cdot s_{\mathsf{t}}$$

$$\leq \exp\left(\frac{2x\eta(\mu\eta - 1)}{\Theta(\mu\eta - 1)}\right) \cdot s_{\mathsf{t}} = e^{\Theta(1)} \cdot \Theta(\mu\eta - 1) = C(\mu\eta - 1),$$

due to $0 \leq r_t \leq \rho_+ \leq \frac{\sqrt{\mu\eta - 1}}{\eta}$ (Lemma 16), and $s_{\mathsf{t}} \leq \Theta(\mu\eta - 1)$ (Lemma 15) where $C > 0$ is some universal constant. Now since all of the facts are true, we obtain the upper bound of the limit $\lim_{t\to\infty} s_t \leq \Theta(\mu\eta - 1)$ and $\lim_{t\to\infty}(r_t, s_t) = (0, s_\infty)$. $\qquad\square$

Finally, we put every piece together and prove Theorem 2.

*Proof.* The proof is almost very similar to the Proof of Theorem 1, and we will omit the identical steps and focus on the difference. When $|x| \in (0, \frac{1}{\eta\mu})$, Lemma 14 suggests that $\lim_{t\to\infty} \alpha_t > 0$. As a result, there always exists a $\mathsf{t}$ such that $\alpha_t > 0$ is true for any $t \geq \mathsf{t}$. As a result, we can use Lemma 17 to show that $\boldsymbol{\beta}_{\boldsymbol{w}_t}$ converges to a linear interpolator as

$$\boldsymbol{\beta}_\infty := \lim_{t\to\infty} \left(p_t^0 - q_t^0\right) \cdot \boldsymbol{\beta}_0 + \left(p_t^1 - q_t^1\right) \cdot \boldsymbol{\beta}_1 + \boldsymbol{\beta}^* = \boldsymbol{\beta}^* + \boldsymbol{\beta}_1 \cdot x^2 \cdot \lim_{t\to\infty} b_t - b_t'.$$

We first show part (1) is true. From Lemma 4, Lemma 5, we can decompose $\boldsymbol{\beta}_{\boldsymbol{w}_t}$ as

$$\boldsymbol{\beta}_{\boldsymbol{w}_t} = \boldsymbol{w}_{t,+}^2 - \boldsymbol{w}_{t,-}^2 = (p_t^0 - q_t^0) \cdot \boldsymbol{\beta}_0 + (p_t^1 - q_t^1) \cdot \boldsymbol{\beta}_1 + \boldsymbol{\beta}_*$$

The convergence speed is then: for any $t \geq \mathsf{t}$:

$$|\langle \boldsymbol{\beta}_{\boldsymbol{w}_t} - \boldsymbol{\beta}_\infty, \boldsymbol{x}\rangle| = |\langle \boldsymbol{\beta}_{\boldsymbol{w}_t} - \boldsymbol{\beta}^*, \boldsymbol{x}\rangle| = |r_t| \leq C \cdot \exp\left(-\Theta(\mu\eta - 1) \cdot (t - \mathsf{t})\right) |r_t|.$$

Moreover, it holds that

$$\|\boldsymbol{\beta}_\infty - \boldsymbol{\beta}^*\| = \lim_{t\to\infty} (p_t^1 - q_t^1) \cdot \|\boldsymbol{\beta}_1\| = (\mu\eta - 1).$$

$\square$

## A.5 PROOF OF PROPOSITION 1

This subsection contains the convergence proof when $\eta\mu > \frac{3\sqrt{2}-2}{2}$. The mechanism and the steps are most similar to the proof of Theorem 2, whereas when $\eta\mu > \frac{3\sqrt{2}-2}{2}$, it is difficult to characterize the $r_t$ sequence via the auxiliary $\rho_t$'s, which is necessary for the estimation of convergence speed. Therefore we put more assumptions and only give an asymptotic result. We begin by finding the fixed point of $(g, h)$ and discuss their stability: intuitively, if a fixed point is (locally) stable, then the nearby trajectory will converge to the point; otherwise, the trajectory will diverge from the point Strogatz (2018); Robinson (2012).

**Lemma 18.** *Suppose that $\eta\mu \in (1, 2)$ and $x \in (0, 1)$. The mapping $(g, h) : \mathbb{R}^2 \to \mathbb{R}^2$ admits fixed points $(r_{1,1}, s_{1,1}) = 0 \times \mathbb{R}$ and $(r_{1,2}, s_{1,2}) = \left(\frac{2}{\eta x}, \frac{(1-x)(2+x\mu\eta)}{x}\right)$. Moreover, $(r_{1,1}, s_{1,1})$ is a stable point when when $s \in (\mu\eta - 1, \mu\eta)$, and an unstable point when $x \notin (\mu\eta - 1, \mu\eta)$. $(r_{1,2}, s_{1,2})$ is unstable regardless of the choice of parameters.*

*Proof.* In the proof of Lemma 13, we already show that $(r_{1,1}, s_{1,1})$ and $(r_{1,2}, s_{1,2})$ are fixed point. It only remains to characterize their properties.

We first consider $(r_{1,1}, s_{1,1})$. The Jacobian matrix of $(g, h)$ is defined as

$$\boldsymbol{J} = \begin{bmatrix} g_r & g_s \\ h_r & h_s \end{bmatrix}$$

where $g_r$, $g_s$ and $h_r$, $h_s$ are first order partial derivatives. We plug $(r_{1,1}, s_{1,1})$ in and compute the eigenvalues of $\boldsymbol{J}$

$$\lambda_{1,1} = 0, \qquad \lambda_{1,2} = 1 - 2\mu\eta + 2s.$$

It is easy to show that when $s \notin (\mu\eta - 1, \mu\eta)$, $|\lambda_{1,2}| > 1$ and hence unstable. Instead, when $s \in (\mu\eta - 1, \mu\eta)$, both $|\lambda_{1,1}| < 1$ and $|\lambda_{1,2}| < 1$, hence $(r_{1,1}, s_{1,1})$ is stable.

We proceed and analyze the stability of $(r_{1,2}, s_{1,2})$. When $x \in [0, 1]$, the sum of two eigenvalues can be lower bounded as

$$\lambda_{2,1} + \lambda_{2,2} = \text{Tr}(\boldsymbol{J}) = 5 + \frac{4}{x^2} - \frac{4}{x} + 2x\mu\eta \geq 5.$$

This implies $\max\{\lambda_{2,1}, \lambda_{2,2}\} \geq \frac{5}{2}$. As a result, $(r_{1,2}, s_{1,2})$ is unstable. $\square$

*Proof of Proposition 1.* We use the same argument in the proof of Lemma 14 to show that $\lim_{t\to\infty} s_t \in (\mu\eta - 1, \mu\eta)$: by our assumption, the $(r_t, s_t)$ iteration does not diverge or becomes chaotic. As a result, it will converge to a stable point or periodic stable orbit. Now due to Lemma 13, when $x \in (0, \frac{1}{\mu\eta})$, $(g, h)$ admits no periodic points when $\alpha_t \leq 0$ (or equivalently $s_t \leq \mu\eta - 1$). Therefore, $\lim_{\alpha\to\infty} \alpha_t > 0$ is true and there exists some $\mathsf{t}$ such that $\alpha_t > 0$ or (or equivalently $s_t > \mu\eta - 1$) holds for any $t \geq \mathsf{t}$. By Lemma 18, $(0, s)$ with $s \in (\mu\eta - 1, \mu\eta)$ are the only stable point, then it holds that $\lim_{t\to\infty} r_t = 0$. This implies $b_t$ or $s_t$ also converges due to its update. Using the identical arguments in the proof of Theorem 2, we reach the conclusion $\square$

## B    PROOFS OF RESULTS IN SECTION 5

*Proof of Lemma 1.* Using update in (3), we compute the expansion of $r_{t+2}$ as

$$r_{t+2} = -(1 - \alpha) \cdot r_{t+1} - r_{t+1}^2$$
$$= (1 - \alpha)^2 \cdot r_t - \alpha r_t^2 - r_{t+1}^2 + (r_t + r_{t+1})(r_t - r_{t+1})$$
$$= (1 - \alpha)^2 \cdot r_t - a r_t r_{t+1} - r_t^2 (r_t - r_{t+1}).$$

If $r_t, r_{t+1}$ have different signs, we obtain the following inequality for any $r_t < 0$:

$$r_{t+2} > (1 - \alpha)^2 \cdot r_t.$$

This implies $|r_{t+2}| < (1 - a)^2 \cdot |r_t|$ for negative $r_t$. Since $r_t$'s are oscillating, we assert that the subsequence of negative $r_t$ converges to zero. Using the update in (3), it suffices to conclude that positive $r_t$'s also converge to zero. □

*Proof of Lemma 2.* We first show that $r_t r_{t+1} < 0$ holds for any $r_t \in [r_-, r_+]$. From (3) we know

$$r_{t+1}/r_t = -1 - a - r_t.$$

Therefore if $r_t > -1 - a$ then $r_t r_{t+1} < 0$. It is easy to verify that for any $a \in [0, 1]$

$$1 + a - \frac{a + \sqrt{a^2 - 4a}}{2} > 1 + a - \frac{a + \sqrt{a^2}}{2} > 0,$$

which suggests $r_- > -1 - a$. Then we prove the sign-alternating part.

We proceed and compute the expansion of $r_{t+2}$ as

$$r_{t+2} = (1 + a)^2 \cdot r_t + a r_t r_{t+1} - r_t^2 (r_t - r_{t+1}).$$

Define the following polynomial (do not confuse with $g$ and $h$ in Appendix A)

$$g(s) = -1 - a - s, \qquad h(s) = -sg(s) \cdot (1 + a + g(s)).$$

To prove that $|r_{t+2}| \geq |r_t|$ when $r_t \in (s_-, s_+)$, it suffices to show that $h(s) \geq s$ when $s \in (s_-, s_+)$. Clearly $h(s)$ is cubic in $s$ with its limit to be $-\infty$ and $\infty$ when $t$ goes to $\infty$ and $-\infty$. Let $s_0$ be the larger stationary point of $h(\cdot)$, it is easy to assert $s_0$ is a local minimum. With MatLab symbolic calculation we verify that $h(s_0) \geq 1$ for any $a > 0$. Then there exists a range such that $h(\cdot) > 1$. It is easy to verify the range is $[s_-, s_+]$ where $s_-$ and $s_+$ are defined as in the statement of lemma. Since $a \in (0, 1)$, it holds that $s_- < 0 < s_+$ and hence finishes the first part of the proof.

To determine the limit of positive and negative subsequence, we assume for simplicity that $r_{2k} < 0$ and $r_{2k+1} > 0$ for any $k \in \mathbb{N}$. Then the limits of both sequences are the solutions to equation $h(s) = s$. From the above discussion we can conclude that the following limits

$$\lim_{k \to \infty} r_{2k} = s_-, \qquad \lim_{k \to \infty} r_{2k+1} = s_+.$$

and finish the proof. □

