# OpenReview forum: "Criteria and Bias of Parameterized Linear Regression under Edge of Stability Regime"
_ICLR.cc/2025/Conference — Submitted to ICLR 2025_

### Official Review · Reviewer_uoKS · 2024-10-23

**Soundness:** 3
**Presentation:** 3
**Contribution:** 3
**Rating:** 6
**Confidence:** 2

**Summary:**

This paper explores the Edge of Stability (EoS) phenomenon. They study GD and focus on a constant step-size exceeding the typical threshold of $2/L$, As the contributions, they observe that EoS occurs even when the loss $l$ is quadratic under proper conditions while the existing works require a subquadratic $l $. Due to the close relationship between the quadratic $l$ and the  depth-2 diagonal linear network, the findings provide some explanations for the implicit bias of diagonal linear networks.

**Strengths:**

The work seems to be solid.

The paper is polished very clearly.

The paper focuses on understanding the convergence of the GD algorithm, breaking through the limitations of traditional smoothness analysis (the step-size exceeds the threshold of $2/L$),  and extending previous conclusions to more complex scenarios (from subquadratic growth objective functions to quadratic growth objective functions).

**Weaknesses:**

Does the conclusion of the more general $n$-sample case in the paper apply to SGD? Due to the importance of stochastic optimization, I expect to see similar conclusions under stochastic optimization as well.

I am sorry I am not very familiar with this topic.  I will revise my score based on the comments of other more senior reviewers.

**Questions:**

What is the behind insight in selecting a specific formula of $\beta$ as $w_+^2-w_-^2$?

Does the theoretical analysis hold in the more general $n$-sample case, or what are the difficulties in this analysis?

---

> ### Author Response · Authors · 2024-12-03
>
> We thank the reviewer for the comments. We will try our best to explain the concerns raised by the reviewer.
>
> **Extension to SGD**
>
> Indeed, EoS phenomena can also occur with SGD [Cohen et al., 2022]. However, most works addressing EoS in stochastic methods are purely empirical, and very few theoretical results are known currently. This is due to the inherent difficulty in establishing convergence for GD on most practical models, and even more so for SGD, which introduces additional complexities. As for the two-layer diagonal neural networks, empirical evidences also suggest the occurance of EoS for SGD. Nevertheless, tackling the theoretical analsyssi remains a very challenging future task.
>
> **Intuition behind quadratic parameterization**
>
> As mentioned in our manuscripts, the study of two-layer diagonal neural networks (DLN) has been emerged as a very important topic in the past several years. This is because (1) DLNs are sufficiently non-linear to capture key properties of more practical deep neural networks, and (2) their simple form allows for a detailed analysis of the GD trajectory. As a result, the two-layer DLN has become one of the most well-studied toy models for implicit bias.  As a result, it becomes one of the most well-studied toy model in terms of implicit bias. The two-layer DLN is equivalent to the linear regression problem where the interpolator/weight vector is forced to admit a quadratic parameterization as $\beta = w_{+}^2 - w_{-}^2$. Therefore, we believe it is a good candidate to study the condition of EoS and convergence of GD under EoS regime.
>
> **Difficulty in theoretical extension to multi-sample setting**
>
> We thank the reviewer for raising this important question. Indeed, we believe that extending the analysis to the $n$-sample case is a highly challenging task—not only for our model but also for related studies, such as [Ahn et al., 2023], [Song and Yun, 2023], among others. More generally, the inability to fully characterize the implicit bias of GD in more complex and more practical models remains a significant limitation of all the existing EoS results. The primary reason behind this is that EoS typically arises in highly non-linear loss landscapes. These objectives often exhibit intriguing geometric properties, making it difficult to track the trajctory of gradient-based methods. As a result, the majority of EoS analysis focused on toy models or the simplication of practical models.
>
> When specializing to our model or the models considered in [Ahn et al., 2023] and [Song and Yun, 2023], the primary challenge in extending the analysis to the $n$-sample setting lies in the difficulty of effectively tracking the residual values across different sample points. As oscillation analysis plays a crucial role in these analysis, the coupling between residuals at different samples under the non-degenerate cases prevents us from obtaining meaningful results. The residuals influence each other through shared parameters, creating a highly complicated landscape that is difficult to decouple or simplify. Therefore, how to tackle the interaction between sample points will be a crucial step in future explorations.

---

### Official Review · Reviewer_VjG6 · 2024-11-03

**Soundness:** 3
**Presentation:** 3
**Contribution:** 2
**Rating:** 5
**Confidence:** 4

**Summary:**

In this paper, the authors consider the task of finding interpolators for the linear regression with quadratic parameterization and study the convergence of constant step-size GD under the large step-size regime. They focus on the non-trivial question of whether a quadratic loss can trigger the Edge of Stability phenomena. The authors show through both empirical and theoretical aspects that, when some certain condition is satisfied, EoS indeed occurs given quadratic loss.

**Strengths:**

In this paper, the authors consider running GD with a large constant step-size to find linear interpolators that admit quadratic parameterization for the one-sample linear regression task. They theoretically proved the on-sample case and extend the one-sample results by empirically finding conditions in the more general n-sample cases. Theses theoretical analysis and empirical results are presented with clear structures.

**Weaknesses:**

The major weakness of this submission is that the authors only provide the theoretical analysis and mathematical proof for the one sample case. The empirical results from the numerical experiments in section 3 shows the convergence of GD under the EoS regime when the loss function is quadratic. The authors only present the theorems characterizing the convergence of GD when the model considered has dimension $d = 2$. The authors should extend these theoretical analysis to the high dimension cases. The one sample analysis and teo dimensional proof is not enough to explain the empirical results from the numerical experiments.

**Questions:**

The authors argued that the model considered in this paper is the depth 2 diagonal linear network. Is that possible to extend the theoretical analysis to the model with higher depths?

---

> ### Author Response · Authors · 2024-12-03
>
> We thank the reviewer for the feedbacks and would like to address the concern raised below.
>
> **Limitation of one-sample and $d=2$.**
>
> We would like to first discuss the first point. Indeed, our current technique does not allow to extend the theoretical analysis to the more general $n$-sample setting. As emphasized in our manuscript, the oscillation of the residual term $r_t$ is very crucial concept in our theoretical analysis. When multiple sample are allowed, the residuals evaluated at different data points become correlated and their mutual influences make the analysis intractable. This coupling of residuals introduces challenges that cannot be easily addressed with standard linear algebra tools, preventing us from fully characterizing the implicit bias of GD under the EoS regime in the multi-sample setting.
>
> Nevertheless, this limitation does not only apply to our results, but also to many important previous works like [Ahn et al., 2023] and [Song and Yun, 2023], among others. More generally, the inability to fully characterize the implicit bias of GD in more complex and more practical models remains a significant limitation of all the existing EoS results. The primary reason behind this is that EoS typically arises in highly non-linear loss landscapes. These objectives often exhibit intriguing geometric properties, making it difficult to track the trajctory of gradient-based methods. As a result, the majority of EoS analysis focused on toy models or the simplication of practical models. Therefore, we believe our work still carries important messages to the EoS community despite the limitions shared by many other works of the same kind.
>
> The second limitation is a restriction to the simple $d = 2$ regime. However, we emphasize that this does not diminish the significance of our current analysis, just as the analysis in [Ahn et al., 2023] remains highly insightful despite its assumptiont it assumes $d=1$. Moreover, in the one-sample case, the empirical behavior and the major characterizations of EoS under both $\eta\mu < 1$ and $\eta\mu > 1$ are consistent for any $d \geq 2$.  Therefore, while linear algebra limitations prevent us from extending the analysis to higher dimensions, our results are still sufficient to capture the essential properties of EoS convergence for GD in the setting we investigate.
>
> **Extension to higher depth.**
>
> We thank the reviewer for raising this fascinating question. We believe the depth will not greatly affect the theoretical analysis in the setting of one-sample and $d=2$, given some minor changes. This is because we can always decompose $\beta_t$, $w_+^d$ or  $w_-^d$ as the span of vectors $\beta_0$, $\beta_1$ and $\beta^*$, and transform the GD dynamics into simple updates of scalars, which are easy to analyze.

---

### Official Review · Reviewer_kgFC · 2024-11-03

**Soundness:** 3
**Presentation:** 3
**Contribution:** 3
**Rating:** 5
**Confidence:** 3

**Summary:**

The paper studies the implicit bias of GD on quadratic loss and linear models with diagonal linear network parameterization under the EoS regime. The paper first empirically shows that when data is in multi-dimension ($d\geq 2$), EoS can occur for quadratic loss, while prior works on EoS suggest that subquadratic loss is necessary for EoS to happen. The experiments show that different choices of step size lead to different oscillation types, from GF regime to EoS regime to chaos to divergence. The paper then theoretically studies the parameter convergence (directly related to generalization) in a sparse solution 2-D diagonal linear network setting and provides non-asymptotic rates in various settings. The results show that in the EoS regime when the step size is not too large ($\eta\mu<1$), smaller initialization ($\alpha$) yields a better generalization, while when step size is large ($\eta\mu>1$) there will be an error floor that cannot vanish as $\alpha\to 0$.

**Strengths:**

1. The paper is overall well-written and easy to follow. It introduces the settings clearly and provides sufficient illustrations to help present results in different conditions/regimes. The main findings are organized clearly. The proof overview is intuitive for grasping the essence of the proof technique.

2. The paper studies implicit bias in the EoS regime for diagonal linear networks in multi-dimensions, which is a significant step forward as most of the prior works on EoS only deal with minimalist examples where the data is assumed to be 1-d, and it is interesting to see the empirical occurrence of EoS depends on the data dimension (non-degeneracy). Moreover, the diagonal linear network setting is closely related to the GD implicit bias literature, so it can potentially connected to wider works.

**Weaknesses:**

My major concern is regarding the "EoS regime". In the paper, there is no rigorous definition of the EoS regime. Instead, a flip sign condition $r_tr_{t+1}<0$ is used throughout the theoretical statements. However, I doubt whether the sign flips in residual indeed correspond to the commonly referred EoS phenomenon ($\eta L_t>2$ and oscillations in loss happen along the trajectory). For example, if we minimize $f(x)=\frac{1}{2}x^2$ with GD, the sharpness is $1$ and the residual $r_t=x_t$ is flipping its sign if we choose step size $\eta\in(1,2)$, but the loss is actually monotonically decreasing and the sharpness is always below $2/\eta$.
In the proof overview (Section 5), it is discussed that $|r_{t+2}|\leq (1-\alpha)^2|r_t|$ is possible, but there is no statement comparing $|r_{t+1}|$ and $|r_t|$, and there is no statement on the relationship between step size $\eta$ and sharpness $L$, so we are not sure if EoS is happening or not. In my understanding, it might be the case that only the $\eta\mu>1$ case corresponds to the EoS that people refer to.

Minor typos (not exactly weaknesses):

1. In Figure 4 the x-axis has no label, which I guess is $\alpha$.

**Questions:**

1. For convergence in the regime $\mu\eta<1$, the current analysis assumes period-2 flip sign in residual, and the result essentially states ${\ell(w_{2t})}_{t\geq 0}$ converges. I am wondering if the analysis can be extended to deal with more general oscillations without specific periods, as they are closer to the EoS phenomenon in practice [1].

2. The paper assumes a specific scaling initialization (lines 218 and 238) for subsequent analysis and the authors claim this initialization is to align with the literature on diagonal linear networks. Regarding this initialization, I have a few questions:
(a) Why in the theory $w_+$ and $w_-$ are of the same scale while in experiment $w_+$ has larger scale than $w_-$? (b) How important is this initialization for ensuring convergence and EoS? (c) How easy is the analysis to be extended to more general or random (e.g. Gaussian) initializations? Just for quick comparison, some existing works can handle general initializations as long as the initial weights satisfy certain conditions [2,3] while they mainly focus on the 1-d case.

3. The result in the $\eta\mu>1$ regime (Figure 4, left) suggests that smaller step size allows smaller generalization errors. This seems to contradict a common belief that a large learning rate is beneficial to generalization as it forces the minimizer to be in a flat region. Could authors provide some insights about this difference?

[1] Cohen, Jeremy M., Simran Kaur, Yuanzhi Li, J. Zico Kolter, and Ameet Talwalkar. "Gradient descent on neural networks typically occurs at the edge of stability." arXiv preprint arXiv:2103.00065 (2021).

[2] Ahn, Kwangjun, Sébastien Bubeck, Sinho Chewi, Yin Tat Lee, Felipe Suarez, and Yi Zhang. "Learning threshold neurons via edge of stability." Advances in Neural Information Processing Systems (2023).

[3] Wang, Yuqing, Zhenghao Xu, Tuo Zhao, and Molei Tao. "Good regularity creates large learning rate implicit biases: edge of stability, balancing, and catapult." arXiv preprint arXiv:2310.17087 (2023).

---

> ### Author Response · Authors · 2024-12-03
>
> We thank the reviewer for their insightful feedback. We will try our best to address the concerns raised, particularly the major issue highlighted in the "Weakness" section. Since this concern is complex, we have broken our response into three parts: Q1, Q2, and Q3.
>
> **Q1. Definition of EoS**
>
> We are sorry for not clearly stating the definition of EoS in the original manuscript. In fact, throughout the whole article, we use EoS in its *original* sense: the phenomena that the sharpness $S_t$, i.e. the largest eigenvalue of objective function's Hessian matrix, crosses the threshold of $2/\eta$, where $\eta$ is the step-size of GD. We have demonstated empirically EoS does happen with the one-sample Diagonal Neural Network model and under the regimes of $\eta\mu<1$ and $\eta\mu>1$ through the experiments and Figure 2 in Section 3. We have added some new lines in the revised version for better clarity on this point.
>
> We understand the reviewer's concern regarding the relationship between EoS and oscillation, and we would like to clarify why our theoretical analysis focuses on the oscillatory behavior rather than the quantity  $2 - \eta S_t$. To illustrate this, it is useful to compare our approach with that of [Ahn et al., 2023], which carefully characterized the trajectory of $S_t$ in its theoretical analysis. This is because the gap between $2 - \eta S_t$ is a key quantity in its "quasi-static" analysis and plays an crucial role in establishing the oscillating convergence of GD. In contrast, our convergence analysis of EoS on the one-sample DLN model does not rely on controlling the gap $2 - \eta S_t$. Therefore, we do not put an emphasis on this in our proof. Consequently, we do not emphasize this quantity in our proof. The purpose of theory part is only to establish convergence of GD under EoS, *given* the fact that the occurance of EoS is already confirmed through the previous empirical section.
>
> This distinction is subtle but important.  Furthermore, in [Ahn et al., 2023], the theoretical analysis only confirmed the existence of oscillation and the fact that $\lim_{t\to\infty}\eta S_t < 2$ by a contrdiction argument. However, the justification for when and where $\eta S_t > 2$ occurs, is based on empirical evidence rather than theoretical proof. In this way, it is very similar to our approach.
>
> Finally, from our theoretical analysis, it is easy to obtain identity $\eta S_t  - 2 = 2\eta( \mu + r_t - c_x \cdot b_t) -2 = 2\eta( \mu - 1+ r_t - c_x \cdot b_t) $ using direct computation. As the result of Theorem 2, we could conclude that $\lim_{t\to\infty} \eta S_t < 2$.
>
> **Q2. $\eta\mu>1$ as the "true" EoS**
>
> We agree with the reviewer that the setting $\eta\mu>1$ represents  the more "authentic" EoS regime. Consequently, the convergence analysis in this regime constitutes the primary focus and major contribution of our work. This focus is not merely because EoS and oscillations are consistently observed under  $\eta\mu>1$, but also because, as discussed in Sections 1 and 4, addressing this regime poses a more significant and challenging task: running GD on our model is equivalent to a parameterized discrete dynamical system, where the parameter varies by time. While $\eta\mu<1$ corresponds to such a system where convergence can be established relatively more easily as in existing work like [Chen et al., 2023], it is extremely difficult to show the convergence of GD under regime $\eta\mu>1$ because a crucial phase transition occurs and transforms a system that initially exhibits divergent behavior into one that converges. To the best of our knowledge, this is the first result to rigorously demonstrate such a kind of convergence.
>
>
> **Q3. Oscillation and EoS**
>
> We agree with the reviewer that oscillation is not necessarily related to EoS for all the regimes. Nevertheless, we believe the focus of the oscillation behavior does not hurt the validity of our result. As we explained in Q1, we stick to the classical definition of EoS and showed empirically that EoS does occur under our setting, which does not depend on the sign-changing of residual $r_t$.
>
> Moreover, we improve Theorem 2 in the latest version and provide a convergence analysis under the more important regime $\eta\mu > 1$ that **no longer** relies on the sign-changing assumption of $r_t$. This is at a cost of restricting $\eta \mu - 1$ to be upper bounded by a universal constant. We believe the removal of the assumption makes our result more convincing in this perpsective. The revisions are marked in red.

---

> > ### Author Response · Authors · 2024-12-03
> >
> > **Below are the replies to the other concerns raised in the "Questions" part.**
> >
> > **Q4. Convergence proof under regime $\mu\eta \leq 1$.**
> >
> > We thank the reviewer for raising this interesting question. In fact, under the regime $\mu\eta - 1$, it is possible to establish the convergence whenever the oscillation occurs or not. We provide below an intuitive explanation. WLOG, suppose $r_t<0$. When $r_{t+1}>0$, the contraction $|r_{t+2}| <  (1 - C_1) |r_t|$ can be established similar to proof in Theorem 1 and the non-positivity of $r_{t+2}$ can be established by a proof similar to Lemma 16. It then suffices to consider the case of $r_{t+1} < 0 $. The update of $r_t$ formulates iteration $$r_{t+1} = - (1 - \alpha_t + \beta_t r_t ).$$
> > When $r_t,r_{t+1}<0$, we upper bound the iteration by plugging the definitions of $\alpha_t$ and $\beta_t$:
> > $$r_{t+1}  \geq  (1 - 2\eta (\mu + r_t - c_x b_t) )\cdot r_t   = (1 - 2\eta (1+x^2) \cdot ( a_t + x^2 \cdot b_t )) \cdot r_t.$$
> > Hence $|r_{t+1}| < (1 - C_2) \cdot |r_t|$ is true since there exists a universal lower bound for $a_t,b_t>0$. This gives a more flexible way of proving convergence under the $\eta\mu<1$ regime.
> >
> > **Q5. Scale of initialization**
> >
> > We are very sorry for the imprecise caption in Fig. 2. In fact, we use initialization $w_{+} = w_{-} = \alpha \mathbf{1}$ for all the experiments, except for run in the 4th column of Fig. 2. The purpose of Fig. 2 is to show that the conditions in Claim 1 are necessary for incurring EoS, and relaxing each one of them will cause it to fail. We made this exception in the 4th column and use unbalanced initialization $w_{+} \neq w_{-}$ because the default initialization will result in $r_t = 0$ for any $t$, which is trivial and is not able to demonstrate the necessity of the condition. We have corrected this in the new version.
> >
> > Extending the current result to other initialization seems to be a rather interesting question. From an empirical perspective, a properly chosen small initialization does not affect the general behavior of EoS, except for the test loss which critically relies on the choice of $\alpha$ due to the sparsity-induced property of diagonal linear network. For the theoretical analysis in the EoS regime with $\eta\mu > 1$ in the same setting of Theorem 2, we believe the scale of initialization, instead of the shape, is important and it will only affect the length of the second interval and has no affect on the linear convergence, because the convergence rate is solely decided by the quantity $\mu\eta - 1$. As for how the scale of initialization affect the convergence, please refer to the newly added Fig. 4 and the answer to Reviewer zwfP, Q2 for a detailed explaination.
> >
> > **Q6. Counterintuitive generalization error**
> >
> > This is also a very interesting question that is worth exploring in future research and we try to brief our idea on this question. Unlike the gradient-flow regime, the (local) mimizer found by GD under EoS is not necessarily flat because the bouncing and oscillation around $x$ clearly destroys the structure. When specialized to the setting of Theorem 2, the Hessian matrix at infinity admits a largest eigenvalue around $2/\eta$ (from Fig. 2 or the above answer to Q1), and a smallest eiganvalue equal to $0$ by easy computation. The ill-conditioned geometrical property might hurt generalization for the minimzer. The only empirical investigation into this problem is founed in [Even et al., 2023], Fig. 1, where the test loss increases dramatically when the step-size is large enough.
> >
> > [Even et al., 2023] (S)GD over Diagonal Linear Networks: Implicit Bias, Large Stepsizes and Edge of Stability. Mathieu Even, Scott Pesme, Suriya Gunasekar, Nicolas Flammarion.

---

### Official Review · Reviewer_zwfP · 2024-11-11

**Soundness:** 4
**Presentation:** 4
**Contribution:** 3
**Rating:** 8
**Confidence:** 3

**Summary:**

The authors analyze the Edge-of-Stability (EoS) phenomenon for gradient descent on linear regression with the loss function $\ell(\langle x, \beta\rangle - y)$, where $(x, y)$ is the datapoint with $x\in \mathbb{R}^d$ and $y\in \mathbb{R}$. EoS phenomenon is observed when GD is run with a stepsize $\eta > \frac{2}{L}$ where $L$ is the smoothness constant of the loss. In the EoS regime, GD oscillates rapidly but still converges to the minima, under certain conditions.

Existing works (Ma et al 2022, Ahn et al 2022, Song & Yun 2023) show that for sub-quadratic $\ell$, GD can enter the EoS regime. This paper shows that for a particular quadratic parameterization, namely diagonal neural networks, GD can enter the EoS regime even for quadratic $\ell$. The parameterization is $\beta = \beta_{w} = w_{+}^2 - w_{-}^2$ and $w= [w_{+}^\top, w_{-}^\top]^\top$, with gradient updates on $w$.


From Claim 1, for quadratic $\ell(s)  = s^2/4$, they obtain EoS for GD under a single-sample regime for $d\geq 2, y\neq 0$ and $x$ non-degenerate. Further, for $d=2, x= (1,x')$ and sparse realizable model $\beta^\star = (\mu, 0), y= \mu$, with initialization scale $\alpha$, they obtain two separate regimes even for EoS.

In the first regime, from Theorem 1, for $\mu \eta <1$ and constant $\alpha$, GD results in both the traditional Gradient Flow regime (GF), without oscillations, and the EoS regime. Here, EoS occurs with damped oscillations. Further, the final solution of GD has generalization error dependent on initialization $\alpha$.

In the second regime, from Theorem 2, $\eta \mu \in (1,2)$, with initialization, $x'$, and generalization error dependent on $\eta\mu$, GD in EoS might initially have diverging amplitude of oscillations, however, it eventually dies down and after a point converges at a linear rate.


Empirically, the authors show that their model requires overparameterization, as for $d=n$, EoS doesn't occur but for $d>n$, it does. For their case of single sample $n=1$, justifying their choice of $d=2$.


**References**--
- (Ma et al 2022) Beyond the Quadratic Approximation: the Multiscale Structure of Neural Network Loss Landscapes. Arxiv.
- (Ahn et al 2023) Learning threshold neurons via the “edge of stability”. NeurIPS.
- (Song & Yun 2023) Trajectory Alignment: Understanding the Edge of
Stability Phenomenon via Bifurcation Theory. NeurIPS.

**Strengths:**

- **Novel insights**: There are several novel and non-trivial insights  -- i) quadratic losses can lead to EoS, ii) the diagonal NN model fulfils this requirement, iii) EoS can have regimes with oscillations with increasing magnitude but still converge later, iv) overparameterization might be necessary for EoS on quadratic losses.

- **EoS on diagonal neural networks**: Diagonal neural networks serve as a simple to analyze but still expressive model. Using these models has 3 important advantages -- i) as these are real networks, their claimed phenomenon occurs not just on some theoretically well-crafted model, ii) diagonal NNs remain a good testbed for theoretical analysis of complicated deep learning phenomena, iii) they have provided a proof for EoS on diagonal neural networks, which was missing from existing works (Even et al 2023), and can now be used to verify claimed empirical phenomenon (Even et al 2023). Note that the proof of EoS on diagonal neural networks is highly non-trivial especially for the $\mu\eta > 1$ case.


- **Presentation**: The paper is easy to read inspite of the heavy notation. The key insights are clearly explained and the figures are very helpful in understanding them. Figure 6, in particular, is a good example to intuitively explain the proof sketch.




**References**--
- (Even et al 2023) (S)GD over Diagonal Linear Networks:
Implicit Bias, Large Stepsizes and Edge of Stability. NeurIPS.

**Weaknesses:**

- **Is subquadratic growth "necessary" or "sufficient" for observing EoS**? It might be beneficial to state the exact reference for the subquadratic condition. Note that the subquadratic condition was introduced in (Ma et al 2022), which the authors have not mentioned. Further, I'm not sure if (Ahn et al 2023) actually state that subquadratic growth is "necessary" for EoS. Assumption A2 and A3 in (Ahn et al 2023), show that subquadratic growth is sufficient for EoS. Similarly, the results in Section 4 in (Ma et al 2022), and assumptions 2.4 and 4.2 in (Song & Yun, 2023) are sufficient for EoS. If I'm missing some details and subquadratic condition is indeed necessary for EoS, can the authors should probably specify the exact theorem, assumption or an argument for this?

- **How large is $\mathfrak{t} - t_0$** ? In Theorem 2, there are $3$ phases for GD. In the first phase, it has not started oscillations, which lasts until $t_0$. From Lemma 7, $t_0 \geq \Omega_{\mu, \eta}(\log(1/\alpha^2))$, but only for $\mu\eta \in (0,2)$, which includes $\mu \in (1, \frac{3\sqrt{2} - 2}{2})$. In the second phase, which lasts from $t_0$ to $\mathfrak{t}$, the oscillations finally start decreasing in magnitude. Lemmas 14 and 15 establish that there exist such a $\mathfrak{t}$, but not how large it is. As after $\mathfrak{t}$, we see linear convergence, how long do we need to wait for it becomes an important question. If the authors cannot establish it theoretically, they might argue empirically that $\mathfrak{t}$ not very large.

- Typo in Line 1537: $|\lambda_{1,2}| < 1$.

**References** --
- (Ma et al 2022) Beyond the Quadratic Approximation: the Multiscale Structure of Neural Network Loss Landscapes. Arxiv.
- (Ahn et al 2023) Learning threshold neurons via the “edge of stability”. NeurIPS.
- (Song & Yun 2023) Trajectory Alignment: Understanding the Edge of
Stability Phenomenon via Bifurcation Theory. NeurIPS.

**Questions:**

- How does the level of overparameterization affect the EoS phenomenon? This might be an easy extension of Figure 5, by checking if increasing the level of overparameterization, i.e., the ratio $\frac{d}{n}$, changes the EoS phenomenon. Figure 5 contains $\frac{d}{n} = 2, 1$, but another experiment on a larger value of $\frac{d}{n}$ might show if large overparameterization helps in EoS. Note that this is not strictly required but might be interesting.

---

> ### Author Response · Authors · 2024-12-03
>
> We sincerely thank the reviewer for the positive feedback and insightful comments. It is evident that the reviewer has a strong understanding of the relevant literature and the core contributions of our work. We appreciate this and kindly request consideration for an increased confidence score. Below, we address the reviewer's specific concerns.
>
> **Q1. Subquadratic being a sufficient condition.**
>
> We apologize for any ambiguity regarding EoS's condition. In previous works such as [Ahn et al 2023], a subquadratic loss is presented as a sufficient condition for EoS, whereas the main message of our empirical and theoretical analysis is to provide some additional necessary conditions to allow EoS to occur even with quadratic loss functions. We have modified some lines to avoid the imprecision.
>
>
> **Q2. How large is $\mathfrak{t}-t_0$.**
>
> We appreciate this important question, which we did not elaborate on in the paper due to multiple reasons. In brief, the length of the second phase, i.e. the gap between $\mathfrak{t}$ and $t_0$, also scales in $\log(1/\alpha)$. Specifically, in the second phase, the residual $r_t$ oscillates and its envelope remains non-shrinking because $\alpha_t < 0$ holds,  leading to an almost linear increase in $b_t$ according to the update $$b_{t+1} = (1 - x\eta r_t)^2 \cdot b_t.$$ The linear increase continues until $b_t$ reaches the scale of $\Theta(\eta\mu - 1)$, at which $\alpha_t$ becomes positive and marks the end of the second phase. From Lemma 7. we know $b_{t_0} \propto \alpha^{\Theta(1)}$. Therefore, we can conclude that the linear increase of $b_t$ lasts for $\log(1/\alpha)$ iterations before phase transition at $t=\mathfrak{t}$.
>
> In the revised manuscript, we have chosen to present this result in Section 3, marked in red, and provide supporting empirical evidence in Figure 4. A detailed theoretical explanation would require additional assumptions in Theorem 2. We think it is more reasonable to keep Theorem 2 with the minimal assumptions needed. Therefore, we believe it is more appropriate to focus on empirical validation in this context.
>
> **Q3. Influence of overparameterization in the EoS phenomenon.**
>
> We thank the reviewer for raising this interesting question. Actually, our empirical evidence indicate that the level of overparameterization, i.e. ratio r = d/n, does not have any significant or consistent influences over the EoS phenomena. This is also true for the cases with $r>2$ which are not covered in the current figures.

---

### Author Response · Authors · 2024-12-03

We sincerely apologize for the late responses before the discussion deadline. We have carefully considered the reviewers' feedback and would like to express our deepest gratitude for the insightful suggestions. Based on their opinions, we have made the following major revisions to the manuscript:

1. As suggested by Reviewer kgFC, we have removed the assumption of sign-changing ($r_t r_{t+1} < 0$) in Theorem 2, which establishes the convergence of GD under EoS and $\eta\mu > 1$, as the major focus of our contribution. This introduces an additional reliance on a universal constant, with $\eta\mu \in (1, \min\{\frac{3\sqrt{2}-2}{2},1+1/C\})$. We believe this would make our results more practical and more self-contained;

2. As suggsted by Reviewer zwfP, we have included a discussion over the duration of second phase under  $\eta\mu > 1$ in Section 3, supported by empirical evidences in Fig. 4.

Additionally, we have corrected typos, grammatical errors, and ambiguous expressions highlighted by the reviewers. All significant revisions are marked in red for clarity.

We would like to remark on the contribution of our work. While the current theoretical analysis is limited by the one-sample restriction and
d=2 setting, we believe our work still offers valuable insights to the study of implicit bias under EoS: the current theoretical analysis of EoS are still far from being complete, with the majority of existing works relying on strong assumptions or impractical carefully-tailored toy models. Therefore, it is more important to focus on the general mechanisms and insights revealed by both empirical and theoretical analysis. In this regard, we identify two major implications of our work:

1. We are among the first to rigorously establish the implicit bias of GD under EoS with a quadratic loss. Moreover, the parameterized linear regression model (i.e., two-layer diagonal linear network) offers a more practical framework compared to other toy models.;

2. The analysis on the more critical regime $\eta\mu>1$ addresses a previously unexplored type of convergence. In contrast to the relative easier regime  $\eta\mu<1$, $\eta\mu>1$ is more challenging from a discrete dynamical system perspective, as it involves a crucial phase transition that transforms an initially divergent system into a convergent one. To the best of our knowledge, this is the first result to rigorously demonstrate such a convergence behavior.

---

### Meta-Review · Area_Chair_zH9g · 2024-12-12

**Metareview:**

This paper investigates EoS phenomena in a simple linear regression problem (quadratic loss) of diagonal linear networks, aiming to challenge prior work suggesting that a subquadratic loss function may be necessary for EoS to occur.

While the reviewers found some of the insights into EoS under quadratic loss interesting and appreciated the clear presentation, the majority believed that the work was limited by the restriction of the theoretical analysis to the simplified one-sample and low-dimensional setting. Although the discussion with the authors partially addressed specific concerns, such as clarifying assumptions and certain theoretical aspects, the broader limitations persist, as highlighted by multiple reviewers. The authors are encouraged to incorporate the important feedback given by the knowledgeable reviewers.

**Additional Comments On Reviewer Discussion:**

Reviewers upheld initial lukewarm assessments.

---

### Decision · Program_Chairs · 2025-01-22

Reject